# Isorhamnetin in Quinoa Whole-Grain Flavonoids Intervenes in Non-Alcoholic Fatty Liver Disease by Modulating Bile Acid Metabolism through Regulation of FXR Expression

**DOI:** 10.3390/foods13193076

**Published:** 2024-09-26

**Authors:** Xiaoqin La, Zhaoyan Zhang, Cunli Dong, Hanqing Li, Xiaoting He, Yurui Kang, Changxin Wu, Zhuoyu Li

**Affiliations:** 1Institutes of Biomedical Sciences, Shanxi University, Taiyuan 030006, China; yanz8816@163.com (Z.Z.); cxw20@sxu.edu.cn (C.W.); 2School of Life Science, Shanxi University, Taiyuan 030006, China; 15735158206@163.com (C.D.); 13903461328@139.com (H.L.); hxt10142022@163.com (X.H.); 18234100882@163.com (Y.K.); 3Shanxi Provincial Key Laboratory of Medical Molecular Cell Biology, Shanxi University, Taiyuan 030006, China; 4Institute of Biotechnology, Shanxi University, Taiyuan 030006, China

**Keywords:** quinoa, isorhamnetin, bile acid metabolism, NAFLD, FXR, BSEP, SLCO1B3

## Abstract

Non-alcoholic fatty liver disease (NAFLD) is a severe hepatic health threat with no effective treatment. Based on the results that Chenopodium quinoa Willd. flavonoids eluted with 30% ethanol (CQWF30) can effectively alleviate NAFLD, this study employed ultrahigh-performance liquid chromatography–electrospray ionization–tandem mass spectrometry (UPLC-ESI-MS/MS) to analyze the components of CQWF30., and screened for flavonoids with potential NAFLD-mitigating effects through network pharmacology. In vitro models using HepG2 and BEL-7402 cell lines induced with free fatty acid (FFA) showed that isorhamnetin administration reduced intracellular lipid deposition and reversed elevated triglyceride (TG) and total cholesterol (T-CHO) levels. In vivo experiments in high-fat diet (HFD) mice demonstrated that isorhamnetin significantly lowered serum and liver fat content, mitigated liver damage, and modulated bile acid metabolism by upregulating FXR and BSEP and downregulating SLCO1B3. Consequently, isorhamnetin shows promise as a treatment for NAFLD due to its lipid-lowering and hepatoprotective activities.

## 1. Introduction

Non-alcoholic fatty liver disease (NAFLD) develops independently of excessive alcohol intake, with an abnormal accumulation of free fatty acid (FFA) and triglyceride (TG) being prominent features [1]. This chronic disease, if left unmanaged, progressively evolves into nonalcoholic steatohepatitis, liver fibrosis, cirrhosis, and even hepatocellular carcinoma, while also amplifying the risk of developing diseases such as diabetes, chronic kidney disease, and heart disease [2]. Over the past four decades, amidst escalating dietary energy intake and sedentary lifestyles, the global prevalence of NAFLD has surged to 37.8% [3], establishing it as one of the most prevalent chronic liver diseases closely associated with liver-related mortality. Consequently, effective intervention in NAFLD is of paramount importance for enhancing overall public health, alleviating the disease burden, and reducing economic losses attributed to healthcare.

The pathogenesis of NAFLD is multifaceted and complex, involving various pathological factors. When the body experiences an overload of FFA, excessive FFA will be transported from the circulation to the liver, leading to fat accumulation within the liver and triggering the development of NAFLD [4]. Under normal conditions, the liver can utilize hepatic lipases to hydrolyze lipids or rely on the action of bile secreted by hepatocytes to reduce the surface tension of fats, facilitating emulsification for the easier digestion of fats and thereby decreasing lipid accumulation [5]. However, in the case of NAFLD, liver damage occurs alongside bile acid stagnation, severely impairing lipid excretion and exacerbating fat accumulation [6]. Studies have shown that bile acids (BAs), the principal components of bile, can reduce liver fat synthesis by suppressing the expression of key genes involved in lipid production [7]. Additionally, these BAs foster the expression of proteins, including farnesoid X receptor (FXR) and Takeda G protein-coupled receptor 5 (TGR5), which collectively stimulate TG metabolism and decrease fat accumulation [8]. FXR, functioning as a pivotal nuclear receptor governing lipid metabolism [9], can inhibit hepatic fatty acid synthesis, accordingly lowering liver TG levels, alleviating hepatic lipid accumulation, and subsequently mitigating the development of NAFLD [10,11].

Moreover, FXR activates the bile salt export pump (BSEP) protein, the primary transporter responsible for BAs secretion from hepatocytes into bile, enhancing the transport and excretion of BAs [12,13]. Activated FXR also suppresses the expression of solute carrier organic anion transporter family member 1B3 (SLCO1B3), reducing the reabsorption of BAs [14]. Paradoxically, the concomitant cholestasis in NAFLD provokes hepatic inflammatory responses, causing liver damage and exacerbating lipid accumulation within the liver. Consequently, the metabolic balance of BAs is crucial to the liver’s normal physiological function. Intervening in bile acid metabolism to decrease its accumulation in the liver and maintain metabolic equilibrium represents a potential approach to treating NAFLD.

The pursuit of novel drugs and therapeutic strategies for NAFLD continues to be a hot topic of research. However, due to the complex pathology and high heterogeneity of disease phenotypes, there is also no specific drug for its treatment. Currently, lifestyle improvement remains the fundamental treatment strategy for NAFLD [15]; however, adherence to such changes is often suboptimal, particularly among severe cases, which has driven the clinical use of interventions such as pioglitazone, aspirin, and statins [2,16]. Nevertheless, these drugs primarily target a single pathway in the disease pathology and have shown less-than-ideal therapeutic effects on NAFLD, with even weaker efficacy in advanced-stage NAFLD patients. In contradistinction, traditional plant-based medicines are esteemed for their milder side effects, and many natural compounds have been proven to exhibit potent hepatoprotective properties. For instance, glycyrrhizin extracted from *Plantago asiatica* L. has been shown to bolster lipid metabolism, showcasing a positive impact on NAFLD [17]. Consequently, identifying natural bioactive components from plants that alleviate NAFLD could pave the way for novel therapeutic strategies for its treatment.

Quinoa (*Chenopodium quinoa* Willd.), a highly nutritious traditional pseudo-cereal [18], is notably rich in flavonoids that exhibit a wide array of pharmacological activities, including antioxidative, anti-inflammatory, and anti-cancer characteristics, and the improvement of metabolic diseases, particularly showing positive regulatory effects on key pathological processes in NAFLD, such as lipid metabolism and insulin resistance [19]. Our previous study highlighted that CQWF30, isolated and purified from quinoa whole-grain, has a remarkable hepatoprotective effect in alleviating NAFLD; nonetheless, the precise identity of the core bioactive constituent remained undetermined [20].

This study employed ultrahigh-performance liquid chromatography–electrospray ionization–tandem mass spectrometry (UPLC-ESI-MS/MS) technology to elucidate the flavonoid components in CQWF30 and, combining network pharmacology, screened flavonoids with the potential to alleviate NAFLD. Through an evaluation of their lipid-modulatory and hepatoprotective efficacies in both in vitro and in vivo models, the efficacious active principles within CQWF30 responsible for its NAFLD-alleviating effects were successfully identified. Furthermore, the mechanisms under the compositions were also uncovered, providing new strategies for NAFLD intervention and scientific grounds for the utilization of quinoa whole-grain as a functional food and liver damage supplement.

## 2. Materials and Methods

### 2.1. Materials

Quinoa whole-grain was sourced from Jingle County (Shanxi, China). D101 macroporous resin (HG 2-885-76) was purchased from Tianjin Bioruns Biotechnology (Tianjin China). Human hepatocellular carcinoma cell lines HepG2 and BEL-7402 were obtained from ATCC (Rockville, MD, USA)). Kaempferol, quercetin, isorhamnetin, oleic acid (OA), a TG assay kit, a T-CHO assay kit, oil red O powder, hematoxylin, Nile red fluorescent dye, 4′,6-diamidino-2-phenylindole (DAPI), optimal cutting temperature (OCT) compound, hematoxylin, and eosin all were acquired from Solarbio (Beijing, China). Fetal bovine serum (FBS), RPMI-1640/DMEM medium, and a CCK-8 assay kit were supplied by Wuhan Prosel Life Science & Technology (Hubei, China). Palmitic acid (PA) was obtained from Merck Chemicals (Shanghai, China). Animal feed (60% fat energy HFD) was obtained from Jiangsu Xietong Pharmaceutical Bio-engineering (Jiangsu, China). Beijing Vital River Laboratory Animal Technology (Beijing, China) supplied 32 male-specific pathogen-free (SPF)-grade C57BL/6N wild-type (WT) mice and the mouse maintenance feed. An LDL-C assay kit, AST assay kit, ALT assay kit, and HDL-C assay kit were all procured from Nanjing Jiancheng Bioengineering Institute (Jiangsu, China). A total bile acid (TBA) content assay kit was obtained from Anhui Baihua Biological Technology (Anhui, China). Simvastatin, the positive lipid-lowering drug used in this experiment, was purchased from Zhejiang Jinhua Pharmaceutical (Zhejiang, China). FXR antibody (25055-1-AP), BSEP antibody (67512-1-Ig), SLCO1B3 antibody (66381-1-Ig), Goat anti-Rabbit IgG (H+L) (SA00001-2), Goat anti-Mouse IgG (H+L) (SA00001-1), and GAPDH antibody (10494-1-AP) were all from Wuhan Mitaka Biotechnology (Wuhan, China). Methanol (67-56-1), acetonitrile (75-05-8), and formic acid (64-18-6) were purchased from Merck KGaA (Merck KGaA, Darmstadt, Germany), and all were LC-MS grade.

### 2.2. Preparation of CQWF30

Quinoa whole-grain was crushed and sieved through a 50-mesh sieve. They were then mixed with a 50% ethanol solution at a solid–liquid ratio of 1:20 (g/mL). The extraction was conducted under ultrasound assistance at 300 W power for 20 min at 55 °C. The mixture was subsequently filtered under reduced pressure to obtain the filtrate, which was further concentrated. After vacuum freeze-drying, the filtrate was resuspended in ultrapure water, and the pH was adjusted to 5.0. It was then filtered through a 0.45 μm membrane to prepare the quinoa flavonoid solution. The purified D101 macroporous resin was washed with 2 BV distilled water, followed by elution with 4 BV volumes of 30% ethanol, with the eluent collected. After concentration under reduced pressure, the product obtained was CQWF30, with a notable effect in alleviating NAFLD. This can be preserved by vacuum freeze-drying for future use.

### 2.3. UPLC-ESI-MS/MS Experiment

A 100 mg sample of CQWF30 was taken and mixed with 3 mm steel beads and 1 mL of 70% methanol. A 70 Hz automated grinder was employed for 3 min to disrupt the mixture, followed by 4 °C ultrasonication at 40 kHz for 10 min. The mixture is then centrifuged at 12,000 rpm for 10 min and then the supernatant is collected. The supernatant was diluted 2–100 times and 10 μL of standard solution (100 μg/mL) were added. After filtering the sample through a 0.22 μm PTFE membrane, it was prepared for UPLC-ESI-MS/MS analysis. The analytical conditions were set as follows: a C18 chromatographic column (Agilent technologies, Santa Clara, CA, USA) is employed with the column temperature maintained at 30 °C and a flow rate of 0.3 mL/min. Mobile phase A consists of water containing 0.1% formic acid, while mobile phase B is pure acetonitrile. The injection volume is 2 μL, with the autosampler kept at 4 °C and a flow rate of 300 μL/min to ensure accurate detection (Appendix A).

### 2.4. Network Pharmacology Analysis

In the PubMed database (https://pubmed.ncbi.nlm.nih.gov/, accessed on 23 December 2021.), a search of the 25 flavonoids contained in CQWF30 and their relation to NAFLD ultimately confirmed 13 flavonoid components. These components were then input into the PubChem database (https://pubchem.ncbi.nlm.nih.gov/, accessed on 13 January 2022.) to obtain their SMILES codes, which were organized and subsequently imported into the SEA database (https://sea.bkslab.org/, accessed on 15 January 2022.) to download their targets. Additionally, through searching databases including GeneCards (https://www.genecards.org/, accessed on 18 January 2022.), TTD (http://db.idrblab.net/ttd/, accessed on 5 March 2022.), OMIM (https://www.omim.org/, accessed on 5 March 2022.), and DrugBank (https://go.drugbank.com/, accessed on 6 March 2022.) using the keyword “NAFLD”, collecting and organizing disease targets related to NAFLD, a Venn diagram to clarify the potential action targets of CQWF30 intervention in NAFLD was constructed by comparing the targets of CQWF30 active components with NAFLD targets using the online tool jevnn (http://jvenn.toulouse.inra.fr/app/example.html, accessed on 8 March 2022.). The interaction network between CQWF30 active components and the potential NAFLD action targets was constructed using Cytoscape software. Furthermore, the target information was imported into the String database (https://cn.string-db.org/, accessed on 12 March 2022.), and exported as a TSV file, which was then used in Cytoscape to build a protein interaction network, with node size and color intensity representing degree values. Subsequently, these potential targets were entered into the Metascape database (https://metascape.org/gp/index.html#/main/step1, accessed on 15 March 2022.), choosing “Homo sapiens” and selecting “GO Molecular Functions”, “GO Biological Processes”, and “GO Cellular Components”, setting the threshold at *p* < 0.05, to conduct Gene Ontology (GO) enrichment analysis and Kyoto encyclopedia of genes and genomes (KEGG) pathway enrichment analysis.

### 2.5. Cell Culture

Human hepatocellular carcinoma cells HepG2 and BEL-7402 were cultured in complete medium (10% FBS and 1% antibiotics) configured with DMEM and RPMI-1640 medium at 5% CO_2_, 37 °C, respectively.

### 2.6. CCK-8 Proliferation Assay

HepG2 and BEL-7402 cells were plated at a density of 3×10³ cells per well into a 96-well plate. After attachment, the culture medium was discarded, and the cells were washed three times with PBS. Subsequently, the cells were subjected to a 4 h treatment with FFA (at a volume ratio of OA to PA 2:1, with 1 mM for HepG2 and 0.5 mM for BEL-7402), to establish a fatty liver cell model. Subsequently, the model cells were treated with kaempferol, quercetin, and isorhamnetin at concentrations of 0, 10, 20, and 40 μM, respectively, with six replicates for each concentration for 24 h. The culture medium was then replaced, and 90 μL of the fresh medium combined with 10 μL of CCK-8 solution was added to each well for a 1 h incubation. Absorbance was measured at 450 nm. The cell proliferation rate was calculated as follows: cell viability (%) = [(As − A0)/(Ac − A0)] × 100%, where As, Ac, and A0 denote the optical density (OD) values of the experimental group, control group, and blank group, respectively.

### 2.7. TG Content and Lipid-Related Indexes Detection Assay

Cell samples were pre-treated according to the procedure above for the CCK-8 assay and 10^4^ cells were collected. The extraction solution was added according to the protocol at a volume ratio of 500:1. Cells were disrupted by ultrasonication on ice, followed by centrifugation at 10,000× *g* for 10 min. The supernatant was carefully collected and kept on ice for T-CHO and TG measurements.

To detect lipid indicators in the blood, the serum collected after centrifugation is used to measure T-CHO, TG, HDL-C, LDL-C, AST, ALT, and AST/ALT ratio according to the protocol.

For liver tissue, homogenization is required, and subsequent steps are the same as for serum sample testing.

For fecal samples collected during the fasting period post-final gavage in mice, a 100 mg portion was taken and mixed with 900 μL anhydrous ethanol at a 1:9 ratio, undergoing homogenization in an ice bath. Following this, the mixture was centrifuged at 2500 rpm for 10 min to isolate the supernatant, which was then further processed according to the instructions.

### 2.8. Oil Red O Staining Experiment

BEL-7402 and HepG2 cells were suspended and seeded at 2 × 10^4^ cells in each well in a 24-well plate (with round coverslip), and the fatty liver model was established as previously described. Cells were then treated for 24 h with kaempferol, quercetin, or isorhamnetin at concentrations identical to those used in the previous experiment. Post treatment, wells were fixed with 0.5 mL of 4% paraformaldehyde at 4 °C overnight, followed by oil red O staining and hematoxylin counterstaining. Samples were examined under a microscope, and digital images were captured to document the experimental results.

### 2.9. Nile Red Staining Experiment

BEL-7402 and HepG2 cells were subjected to Nile red staining and incubated with Nile red fluorescent dye for 10 min at room temperature, after which they were washed three times with PBS to discard the excess stain. Following this, DAPI solution was added to each well to stain the nuclei for 15 min at room temperature, and the samples were covered with an antifade mounting medium. Finally, image acquisition and data analysis were performed using a DeltaVision imaging system.

### 2.10. Animal Experiments

#### 2.10.1. Animal Experiment Design

A total of 32 male, SPF-grade C57BL/6N WT mice were acclimated for one week and then randomly allocated into four groups. The control group continued on a standard diet, whereas the other three groups were shifted to a 60% HFD for 17 weeks to induce a NAFLD model. During the experimental period, the control and HFD groups received equal daily saline gavage, meanwhile, the isorhamnetin (HFD + isorhamnetin) and simvastatin (HFD + simvastatin) groups received a daily gavage of quercetin and simvastatin at 5 mg/kg, respectively, for 16 weeks. Mouse body weights and food consumption were recorded every two days. All of the animal experiments were carried out in strict accordance with the ethical regulations and protocols approved by Shanxi University’s Institutional Animal Care and Use Committee. Ethical code: SXULL2022101, approval date: 8 May 2022, name of ethics committee: Shanxi University Scientific Research Ethics Review Committee.

#### 2.10.2. Hematoxylin and Eosin (H&E) Staining Experiment

Fresh liver tissue samples were fixed in 10% paraformaldehyde solution for 24 h and then rinsed thoroughly with running water until no odor remained. Next, the samples were transferred to a 30% sucrose-PBS solution for osmotic treatment. Subsequently, the samples were embedded with an OCT compound, frozen at −25 °C, and sectioned at a thickness of 8 μm. Then, the staining was performed according to the H&E staining protocol, and the slices were examined under a microscope. Photographic documentation and analysis of the histopathological features of the liver tissue were conducted.

#### 2.10.3. TBA Content Detection Assay

TBA content was measured in mouse serum according to the kit instructions. Liver and fecal serum samples should be homogenized to obtain a supernatant for analysis.

#### 2.10.4. Bile Acid Metabolic Profiling

Take a 20 mg sample of liver tissue and add 10 μL standard solution (Glycochenodeoxycholic Acid-d4) (1 μg/mL), followed by 200 μL 20% methanol-acetonitrile. Homogenize the mixture thoroughly. Then, centrifuge the mixture at 4 °C, 12,000 r/min for 10 min, and collect and concentrate the supernatant. Thereafter, resuspend the sample in 100 μL of 50% methanol–water solution to prepare it for liquid chromatograph–mass spectrometer (LC-MS/MS) analysis. 

#### 2.10.5. Principal Component Analysis (PCA) 

Changes of 20 BAs in the livers of different groups of mice detected by LC-MS/MS were then analyzed through PCA. These 20 baseline variables underwent statistical dimensionality reduction and an appropriate number of principal components with the highest eigenvalues were extracted based on the eigenvalues and cumulative contribution rates. Then, the principal component scores for each bile acid were calculated using the component matrix and eigenvalues, and a comprehensive evaluation and ranking were performed based on these scores.

### 2.11. Western Blot Assay

Cells were processed and collected according to the above steps. WB/IP cell lysis buffer (PMSF: cell lysate = 1:100) was added to the cells, lysed on ice for 30 min, and centrifuged for 15 min at 4 °C, 12,000 rpm, and the resultant supernatant was the total cellular protein. After protein quantification, SDS-PAGE electrophoresis was performed, and specific antibodies FXR, BSEP, SLCO1B3, and GAPDH were applied to label the target proteins, respectively. Finally, the blotting results were assessed and analyzed using ImageJ software. For the extraction of total liver tissue protein, homogenization was performed first.

### 2.12. Statistical Analysis

All datasets were derived from a minimum of three independent biological replicates. Statistical analyses were conducted using SPSS 22.0 software, and differences were considered statistically significant when the *p*-value was less than 0.05. Furthermore, graph generation was assessed via Student’s *t*-test utilizing Origin 2018 software. 

## 3. Results

### 3.1. The Prediction of Active Components and Mechanism (s) in CQWF30 Intervenes with NAFLD

The preliminary efforts in our laboratory have successfully extracted and isolated *Chenopodium quinoa* Willd. flavonoids (CQWF) from quinoa whole-grain, further pinpointing the efficacious component, CQWF30, in mitigating NAFLD [20]. Expanding upon this breakthrough, we employed advanced UPLC-ESI-MS/MS technology to analyze the compositional elements of CQWF30, unearthing 25 flavonoids exhibiting potential bioactive potential (Appendix A). Subsequently, associative searches in Pubmed for these 25 constituents related to NAFLD (Appendix A), revealed a significant correlation for 13 of these flavonoid components (Table 1). This led us to delve into the SEA database, where we uncovered a network of 153 prospective therapeutic targets for these 13 flavonoids, which were then compared and analyzed against 2459 NAFLD-related biological targets obtained from GeneCard, TTD, OMIM, and DrugBank databases, yielding 53 intersecting targets as the potential action targets of CQWF30 in NAFLD intervention (Figure 1A). By constructing an interaction network diagram between the active components of CQWF30 and NAFLD targets, we vividly illustrated the complex interactions among the 13 flavonoid components and the 53 targets, with kaempferol (involved with 37 targets), quercetin (31 targets), and isorhamnetin (24 targets) ranking as the top three active ingredients in terms of target relationship density (Figure 1B).

The study conducted an in-depth exploration of 53 potential targets through GO enrichment analysis. Firstly, by analyzing the biological processes associated with these targets, we found that they primarily participate in reactions involving inorganic substances and peptides, the positive regulation of phosphorylation processes, and the regulation of hormone levels (Figure 1C). In the cellular component (CC) analysis, it was evident that these targets show significant associations with functional areas such as membrane rafts, vesicle lumens, extracellular matrix, glutamatergic synapses, and RNA polymerase II transcription regulatory complexes (Figure 1D). Furthermore, the enrichment analysis of genes related to molecular functions revealed a close relationship between these targets and enzymatic activities, such as oxidoreductase activity and nitric oxide synthase regulatory activity (Figure 1E). Additional pathway analysis using the KEGG highlighted the involvement of these targets in multiple disease-related metabolic pathways. These pathways encompass cancer-signaling pathways relevant to tumorigenesis and development, lipid metabolism and atherosclerosis pathways that demonstrate the link between abnormal lipid metabolism and cardiovascular diseases, as well as bile acid metabolism pathways (Figure 1F). Given the intimate connection between lipid metabolism and bile acid metabolism, the latter being a critical process for regulating hepatic lipid excretion and maintaining bile flow, its dysfunction has been linked to NAFLD [21]. Therefore, it is plausible that the bioactive molecules in CQWF30—kaempferol, quercetin, and isorhamnetin—may intervene in NAFLD by influencing the bile acid metabolism pathway.

### 3.2. Isorhamnetin Can Inhibit Lipid Accumulation in High-Fat Cells Induced by FFA

Based on the network pharmacology screening mentioned above, kaempferol, quercetin, and isorhamnetin were identified as the most likely active components in CQWF30 for intervening in NAFLD. We proceeded to conduct CCK-8 assays on the HepG2 and BEL-7402 liver cancer cell lines, which demonstrated that these constituents had negligible impacts, neither suppressing nor stimulating the proliferative capabilities of these cells (Figure 2A–F).

To delve deeper into the effects of these compounds on alleviating NAFLD, we established a high-lipid cellular model induced by FFA. In HepG2 cells, the FFA-induced model group exhibited a significantly increased accumulation of lipid droplets (Figure 2G–I). However, after treatment with quercetin and isorhamnetin, there was an observed reduction in intracellular lipid droplets (Figure 2H–I). In contrast, kaempferol treatment did not bring about any notable changes (Figure 2G). Similar observations were corroborated in BEL-7402 cells (Figure 2J–L). Subsequently, Nile red fluorescence staining conducted in HepG2 cells demonstrated that both quercetin and isorhamnetin could effectively reduce the accumulation of lipid droplets induced by fatty acids, with isorhamnetin showing particularly significant results, consistent with oil red O staining (Figure 2M–O). The Nile red staining experiment in BEL-7402 cells also highlighted the superior effect of isorhamnetin in reducing lipid accumulation (Figure 2P–R). Based on these findings, we chose to focus subsequent research efforts on isorhamnetin. We performed a quantitative analyses of TG and T-CHO in the FFA-induced NAFLD cellular models. The results indicated that isorhamnetin could effectively inhibit the increase in T-CHO and TG levels induced by FFA in HepG2 cells (Figure 2S,T), a phenomenon also observed in BEL-7402 cells (Figure 2U,V).

Taken together, our study demonstrated that isorhamnetin can inhibit the accumulation of lipids in high-lipid cells induced by FFA.

### 3.3. Isorhamnetin Derived from Quercetin Modulates the Weight of Mice on an HFD and Reduces Fat Accumulation 

Isorhamnetin, a flavonoid compound abundant in fruits and vegetables, is a natural metabolic byproduct of quercetin, generated through metabolic processes in organisms [22,23]. Studies have elucidated that ingested quercetin undergoes gut microbial metabolism, interacting with water to form quercetin-3-β-D-glucoside, which is subsequently hydrolyzed under the mediation of β-glucosidase to ultimately produce isorhamnetin, a physiological conversion that is up to 80% efficient [22]. Additionally, quercetin is also metabolized into isorhamnetin within the liver. Given this, coupled with the significantly greater economic viability of quercetin over isorhamnetin, we opted to use quercetin in replete of isorhamnetin to verify its intervention effects on NAFLD mice [24]. Figure 3A illustrates the experimental design. The results showed that the body weights of mice in the HFD group were notably higher than those of the control group, with substantial increases observed, whereas the isorhamnetin and simvastatin intervention groups significantly inhibited excessive body weight gain in mice (Figure 3B–D). 

Comparing food intake among groups showed no changes, indicating that isorhamnetin did not control body weight in HFD mice by regulating their food intake (Figure 3E). Fundamentally, obesity is characterized by the excessive accumulation of TG in white adipocytes, leading to an abnormal enlargement of fat cells. The anatomical and analytical results showed that the white adipose tissue in the inguinal and epididymal regions of mice in the HFD group exhibited significant increases in both volume and mass compared to the control group, with the fat tissues presenting a visibly swollen state (Figure 3F–H). In contrast, mice in the HFD + isorhamnetin group and the HFD + simvastatin group experienced a notable reduction in fat tissue weight (Figure 3F–H). These findings collectively highlighted the positive role of isorhamnetin in mitigating weight gain induced by an HFD.

### 3.4. Isorhamnetin Intervention Alleviates HFD-Induced Liver Injury and Improves Related Indices in Mice

Following a macroscopic examination of the livers, we initially assessed the damaged condition of mouse livers. The control group exhibited the following typical characteristics of healthy livers: a bright coloration, a smooth and flawless surface, an appropriate texture, and well-defined margins (Figure 4A). In contrast, livers from the HFD group showed obvious swelling, a yellowish discoloration, a slightly rough and greasy texture, increased tissue hardness, and blurred edges, indicating marked pathological changes (Figure 4A). Mice that were administered HFD alongside isorhamnetin or simvastatin also presented livers with reduced yellowing, marginally improved surface smoothness, lessened greasy sensation, and slightly better-defined edges, suggesting ameliorative effects (Figure 4A). In addition, we quantified the degree of liver injury by calculating the liver index (liver wet weight/body weight × 100%). The results showed that both isorhamnetin and simvastatin interventions mitigated excessive liver weight gain induced by the HFD and reduced liver indices, affirming their beneficial roles in alleviating HFD-induced liver injuries (Figure 4B,C).

Histopathological scrutiny through H&E staining revealed that the livers of mice in the HFD group showed severe morphological alterations such as the regional swelling of hepatocytes, cytoplasmic laxity accompanied by edema, as well as the steatosis and fibrosis of the confluent area (Figure 4D). However, the intervention of isorhamnetin significantly reversed these pathological features (Figure 4D). 

These histological observations were corroborated by blood biochemistry, which demonstrated a substantial escalation in T-CHO, TG, LDL-C, AST, and ALT levels among HFD mice, accompanied by a decline in HDL-C. These abnormalities in lipid and liver function indices were effectively corrected following the administration of isorhamnetin (Figure 4E–J). Of note, the De Ritis Ratio (AST/ALT) in the isorhamnetin group was significantly reduced compared to the HFD group, powerfully endorsing isorhamnetin’s efficacy in mitigating NAFLD (Figure 4K). Liver tissue biochemical assays echoed these findings (Figure 4L–R).

Further analysis of the TG content in fecal samples from the mice did not reveal any substantial intergroup variations, implying that the hypolipidemic mechanism of isorhamnetin may not primarily involve enhanced fecal lipid elimination (Figure 4S). Importantly, all mice treated with isorhamnetin did not show any acute gastrointestinal adverse effects throughout the experimental period, underscoring the potential of isorhamnetin as a mild but effective lipid-lowering therapy.

### 3.5. Isorhamnetin Alleviates NAFLD by Reducing Hepatic Bile Acid Accumulation

The dysregulation of lipid metabolism is often closely associated with abnormalities in BA metabolism, and the excessive accumulation of lipids in the liver is a core trigger for the development of NAFLD [25]. Combined with previous network pharmacology findings, modulating the bile acid metabolic pathway has emerged as a new focal point in NAFLD treatment strategies [26].

To explore the underlying influence of isorhamnetin on the bile acid metabolism in the NAFLD model, we employed metabolomics to analyze the concentrations of TBA in serum, liver tissues, and fecal samples of mice. As illustrated in Figure 5A–C, the HFD regimen led to a substantial escalation in TBA accumulation, alleviated by the administration of isorhamnetin. Further using the LC-MS/MS technique, we accurately measured the types and contents of BAs in the livers of each group. 

PCA results revealed tight clustering within groups and distinct inter-group variations in bile acid metabolism, particularly the evident separation between the control and HFD groups, validating the successful establishment of the NAFLD model (Figure 5D). Samples treated with isorhamnetin or simvastatin clustered closer to the control group, suggesting the effective restoration of BA levels towards normalcy (Figure 5D). Expanding on the bile acid metabolic profiles, our analysis encompassed 20 distinct BAs, including 11 primary BAs and their taurine conjugates, 6 secondary BAs alongside their taurine derivatives, and 3 individual secondary BAs (Table 2).

Through box plots and Z-score heatmaps of these 20 BAs, marked fluctuations in liver BA composition induced by the HFD were evident, while the intervention of isorhamnetin effectively adjusted this metabolic imbalance (Figure 5E–Y). In summary, isorhamnetin intervention effectively corrects the bile acid metabolic abnormalities triggered by an HFD, providing new scientific grounds for the treatment of NAFLD.

### 3.6. Isorhamnetin Regulates Bile Salt Metabolism in the Liver by Activating FXR to Alleviate NAFLD

Studies have shown that CDCA is the most efficient endogenous agonist of FXR, while tauro-β-muricholic acid (Tβ-MCA) and ursodeoxycholic acid (UDCA) act as endogenous antagonists of FXR [27]. In light of our experimental findings, where intervention with isorhamnetin led to a notable upregulation of CDCA levels and a clear downregulation of both Tβ-MCA and UDCA (Figure 5Q,M,W), we speculate that the mechanism of isorhamnetin may involve restoring the bile acid metabolic imbalance induced by HFD, thereby subsequently activating FXR activity, both promoting the exocytosis of BAs and restricting their reabsorption process.

To further validate this hypothesis, we examined the expression of FXR, BSEP, and SLCO1B3 in HepG2 cells, BEL-7402 cells, and mouse liver tissues. The results indicated that the expression levels of FXR and BSEP were significantly reduced, while SLCO1B3 was elevated in the HFD group compared to the control. However, after the intervention of isorhamnetin, the expression levels of FXR and BSEP significantly rebounded and the SLCO1B3 level was significantly decreased (Figure 6A–F).

Collectively, these findings demonstrate that isorhamnetin, by activating FXR, not only enhances the expression of its downstream effector BSEP, promoting the discharge of BAs from the liver into bile, but also effectively reduces the reabsorption of BAs by suppressing SLCO1B3 expression, thus effectively alleviating the accumulation of BAs in the liver.

## 4. Discussion

This study characterized the components of quinoa whole-grain flavonoids—CQWF30, and 25 distinct flavonoid compounds were obtained. The subsequent screening of these compounds for correlations with NAFLD pinpointed kaempferol, quercetin, and isorhamnetin as the most intricately associated active ingredients with therapeutic targets, suggesting their potential as active substances in CQWF30 intervention for NAFLD. In vitro experimentation using a high-fat cellular model showcased the lipid-inhibitory effects of quercetin and isorhamnetin, with isorhamnetin exhibiting superior efficacy. Considering isorhamnetin is a metabolic product of quercetin in vivo, it is posited as the paramount constituent within CQWF30 for NAFLD intervention. Further, in vivo experiments revealed notable lipid-lowering effects of isorhamnetin, alleviating liver damage induced by an HFD.

The accumulation of lipids within the liver constitutes a cardinal feature in the onset of NAFLD [25], a process frequently triggered by abnormalities in bile acid metabolism. Elevated bile acid levels in the liver can lead to cytotoxicity, stimulating the production of pro-inflammatory cytokines [11], exacerbating liver injury, and further worsening bile acid stasis, thereby promoting NAFLD progression [12]. Therefore, modulating bile acid metabolism is seen as a potential new strategy for NAFLD treatment [26]. Strategies targeting bile acid reduction encompass the augmentation of bile excretion and suppressing reabsorption [28]. 

We understand that in patients with NAFLD, compared to healthy individuals, the ratio of primary to secondary BAs within the liver can shift, leading to an increased proportion of secondary BAs and a concomitant rise in total bile acid levels, a phenomenon known as bile acid retention. Additionally, the signaling pathways involving BAs may be disrupted, preventing the effective activation of receptors such as FXR, which are crucial for metabolic regulation. In our study, the administration of isorhamnetin significantly reduced the levels of bile acids such as α-MCA, UDCA, ω-MCA, LCA, Tω-MCA, THDCA, and Tβ-MCA in the livers of NAFLD model mice, while concurrently increasing the concentrations of CDCA and TCA (Figure 5). Notably, CDCA is the strongest endogenous agonist of FXR, while UDCA and T-β-MCA act as FXR antagonists. FXR, expressed in the ileum and liver, plays a pivotal role in bile acid homeostasis [13], reducing BA reabsorption, and enhancing BA excretion by the upregulation of BSEP, the major transporter for BA secretion from hepatocytes into bile [29]. Additionally, activated FXR could decrease SLCO1B3 expression, inhibiting BA reabsorption in the liver [29]. Overall, isorhamnetin intervention through modulating the composition of BAs significantly upregulated FXR and BSEP protein expression in the liver, while notably downregulating SLCO1B3 expression (Figure 5E–G), thereby enhancing BA export from the liver to the intestine and reducing BAs reabsorption, improving bile acid metabolic balance and alleviating cholestasis, which enabled the liver to metabolize excess T-CHO, effectively slowing down the progression of NAFLD.

Furthermore, what we have currently observed is that isorhamnetin intervention modulates the bile acid composition within the liver of an NAFLD model; however, the underlying mechanisms of this phenomenon remain unclear. Additionally, it is yet to be determined whether other flavonoids in CQWF30 or the synergistic effects of multiple flavonoid compounds also exert similar interventions on NAFLD. These questions will be the focus of our future research efforts. Our study laid an important scientific foundation for novel NAFLD therapeutics and expanded our understanding of the mechanisms by which plant flavonoids operate in metabolic disorders. Future studies should delve into the human applicability and safety profiles of isorhamnetin, with the anticipation of broadening the therapeutic strategy available for NAFLD management.

## 5. Conclusions

This study utilized UPLC-ESI-MS/MS to identify the components of CQWF30. Through network pharmacology screening and in vitro and in vivo experimental validation, it was discovered that isorhamnetin can regulate bile acid metabolism by upregulating FXR and BSEP and downregulating SLCO1B3, thereby alleviating bile acid accumulation in the liver of NAFLD, thus playing a role in mitigating NAFLD. This finding provides a theoretical basis for dietary interventions in NAFLD.

## Figures and Tables

**Figure 1 foods-13-03076-f001:**
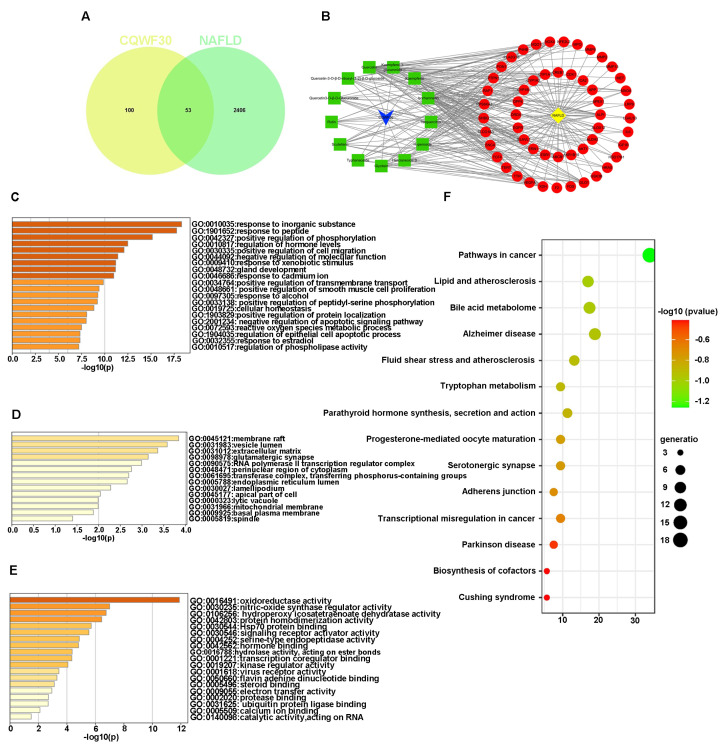
Identification of CQWF30 components and screening of components relevant to NAFLD intervention. Venn diagram illustrating the potential targets of CQWF30 in NAFLD intervention (**A**). The protein-protein interaction (PPI) network of 53 potential targets modulated by 13 flavonoids in CQWF30 in the context of NAFLD (**B**). Gene Ontology (GO) enrichment analysis of biological processes for the potential targets (**C**). GO enrichment analysis of cellular components for the potential targets (**D**). GO enrichment analysis of molecular functions for the potential targets (**E**). KEGG pathway enrichment analysis for the potential targets (**F**).

**Figure 2 foods-13-03076-f002:**
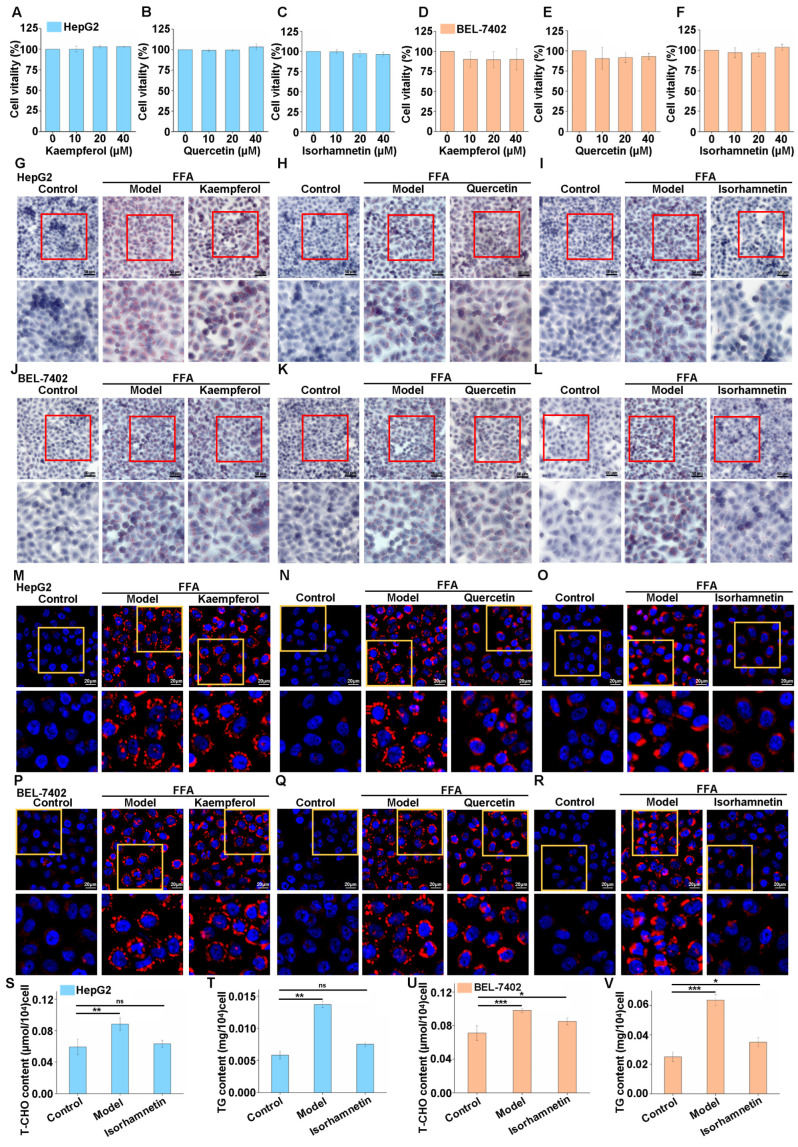
Effects of isorhamnetin on lipid accumulation and cell viability. Effects of kaempferol (**A**), quercetin (**B**), and isorhamnetin (**C**) on the viability of HepG2 cells and effects of kaempferol (**D**), quercetin (**E**), and isorhamnetin (**F**) on the viability of BEL-7402 cells. Oil red O staining and Nile red fluorescence staining showing the effect of 10 μM kaempferol, quercetin, and isorhamnetin on lipid droplets in FFA-induced (at a volume ratio of OA to PA 2:1, 1 mM for HepG2) HepG2 cells (**G**–**I**,**M**–**O**). Oil red O staining and Nile red fluorescence staining showing the effect of 10 μM kaempferol, quercetin, and isorhamnetin on lipid droplets in FFA-induced (0.5 mM for BEL-7402) BEL-7402 cells (**J**–**L**,**P**–**R**). The scale bar for Oil red O staining is 50 μm and 20 μm for Nile red fluorescence staining. T-CHO and TG content in HepG2 cells (**S**,**T**). T-CHO and TG content in BEL-7402 cells (**U**,**V**). The images below are enlarged views of the red or yellow frames in the upper image. * *p* < 0.05, ** *p* < 0.01, *** *p* < 0.001, ns, not statistically significant.

**Figure 3 foods-13-03076-f003:**
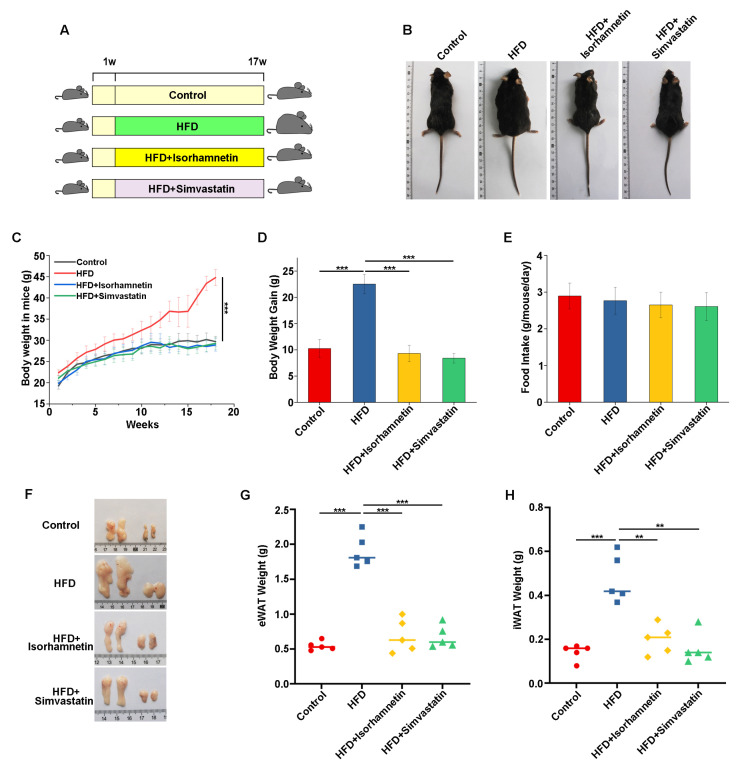
Isorhamnetin reduces lipid accumulation in NAFLD mice. Schematic representation of the animal experiment design and grouping (**A**). Representative morphological images of mice from each group (**B**). Growth curve of body weight (**C**). Body weight gain of mice in each group (**D**). Average daily food intake (**E**). Representative images of epididymal and inguinal white adipose tissues (**F**). Weight of inguinal white adipose tissue (**G**). Weight of epididymal white adipose tissue (**H**). In images (**G**) and (**H**), red indicates the Control group, blue indicates the HFD group, yellow indicates the HFD + Isorhamnetin group, and green indicates the HFD + Simvastatin group. ** *p* < 0.01, *** *p* < 0.001.

**Figure 4 foods-13-03076-f004:**
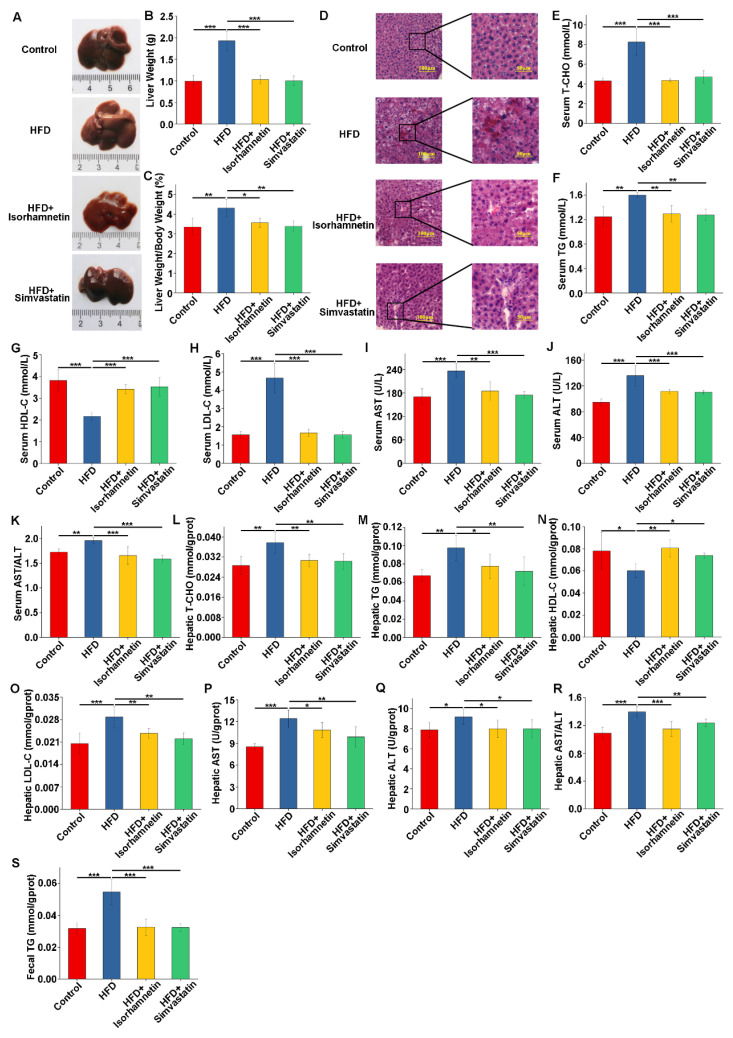
Isorhamnetin alleviates liver morphology and injury and regulates lipid levels in the liver and serum of mice. Representative histological images of liver tissues (**A**). Liver weight (**B**). Liver index (liver-to-body weight ratio) (**C**). H&E staining of liver tissues (**D**). Serum T-CHO (**E**), TG (**F**), HDL-C (**G**), LDL-C (**H**), AST (**I**), and ALT (**J**) levels, and AST/ALT ratio (**K**) in serum. Liver T-CHO (**L**), TG (**M**), HDL-C (**N**), LDL-C (**O**), AST (**P**), and ALT (**Q**) levels, and AST/ALT ratio (**R**) in liver tissues. TG content in feces (**S**). * *p* < 0.05, ** *p* < 0.01, *** *p* < 0.001.

**Figure 5 foods-13-03076-f005:**
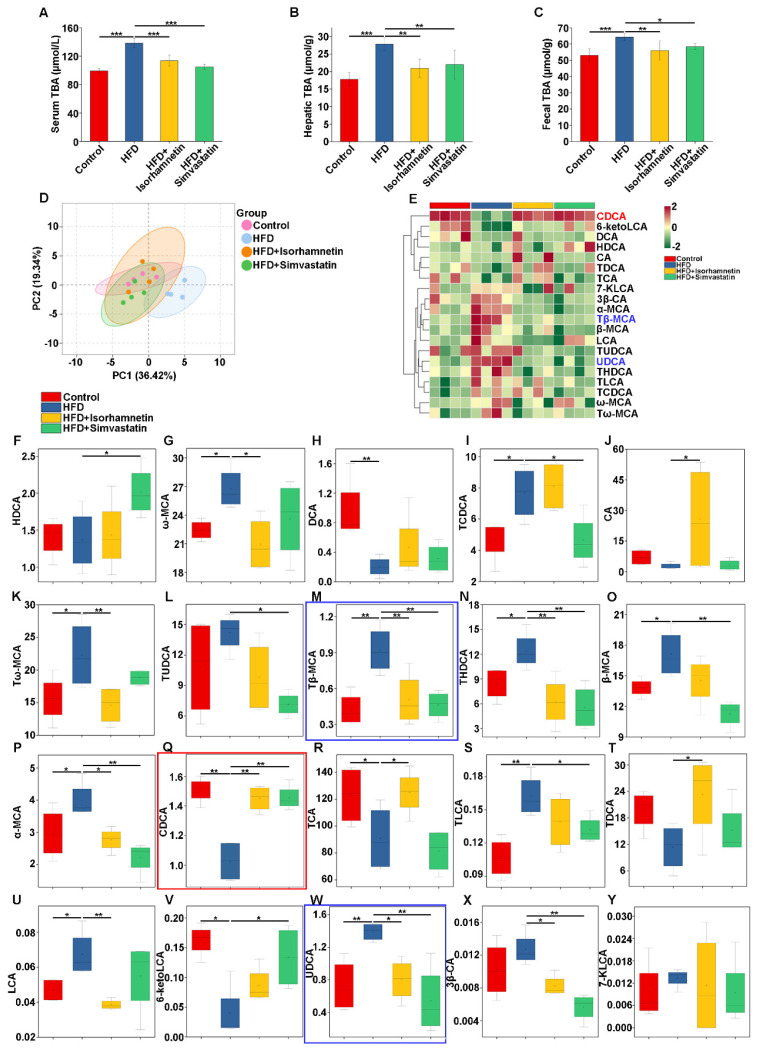
Isorhamnetin mitigates NAFLD by reducing bile acid stasis. Effect of isoquercitrin on TBA in serum (**A**), liver (**B**), and feces (**C**) of C57BL/6N mice. Two-dimensional principal component analysis–discriminant analysis (PCA-DA) score plot (**D**). Heatmap of 20 BAs with a significant contribution to differentiation, where green indicates low Z-scores and red indicates high Z-scores (**E**). Concentrations of various BAs such as Tβ-MCA (**M**), CDCA (**Q**), and UDCA (**W**) in the liver (**F**–**Y**). * *p* < 0.05, ** *p* < 0.01, *** *p* < 0.001.

**Figure 6 foods-13-03076-f006:**
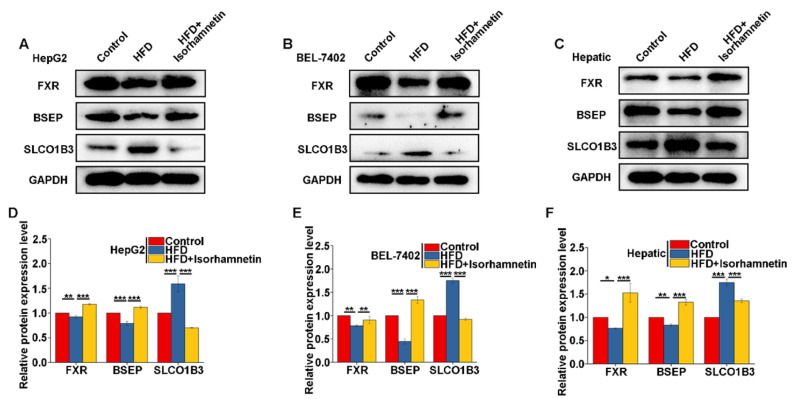
Isorhamnetin relieves NAFLD by increasing FXR protein expression and regulating downstream proteins. Expression of FXR, BSEP, and SLCO1B3 proteins in HepG2 cells (**A**) and FXR, BSEP, SLCO1B3 proteins expressed in BEL-7402 cells (**B**). Expression of FXR, BSEP, and SLCO1B3 proteins in mouse liver (**C**). Quantification of protein expression for FXR (**D**), BSEP (**E**), and SLCO1B3 (**F**) * *p* < 0.05, ** *p* < 0.01, *** *p* < 0.001.

**Table 1 foods-13-03076-t001:** Flavonoids associated with NAFLD in CQWF30.

No.	RT [min]	Formula	Class of Compound	Name	MW	MS/MS Fragment	Percentage of CQWF30
1	6.01	C15H10O7	Flavonoids	Quercetin	302	303.0487	10.54
2	6.23	C15H10O6	Flavanoids	Kaempferol	286	287.05435	8.83
3	6.26	C27H30O15	Flavanoids	Nictoflorin	594	287.05447	7
4	6.49	C27H30O16	Flavanoids	Rutin	610	303.04941	3.84
5	6.29	C34H42O20	Flavonoids	Typhaneoside	770	317.06506	3.43
6	7.10	C21H18O12	Flavonoids	Scutellarin	462	287.05441	2.73
7	6.01	C21H20O12	Flavanoids	Hyperoside	464	303.04947	2.15
8	6.23	C21H20O11	Flavanoids	Orientin	448	449.10721	1.91
9	6.29	C16H12O7	Flavonoids	Isorhamnetin	316	317.06451	0.82
10	7.12	C22H22O12	Flavanoids	Isorhamnetin 3-galactoside	478	301.03491	0.24
11	9.13	C16H12O5	Flavanoids	Glycitein	284	268.03723	0.08
12	2.97	C8H7NO2	Flavanoids	4-Hydroxymandelonitrile	149	150.05487	0.08
13	11.33	C16H12O5	Flavanoids	5-*O*-Methylgenistein	284	285.07532	0.01

**Table 2 foods-13-03076-t002:** The content of 20 BAs in the liver (ng/g).

Index	Compounds	Control	HFD	HFD+ Isorhamnetin	HFD+ Simvastatin
β-MCA	β-muricho lic acid	13.84 ± 0.94	17.11 ± 2.36 #	14.53 ± 2.43	11.28 ± 1.32 **
6-ketoLCA	5-β-Cholanic Acid-3α-ol-6-one	0.16 ± 0.03	0.04 ± 0.05 #	0.09 ± 0.03	0.13 ± 0.05 *
α-MCA	α-muricho lic acid	2.97 ± 0.78	4.00 ± 0.56 #	2.77 ± 0.37 *	2.22 ± 0.51 **
UDCA	Ursodeoxycho lic acid	0.73 ± 0.32	1.39 ± 0.11 ##	0.80 ± 0.26 *	0.54 ± 0.42 **
ω-MCA	ω-muricho lic acid	22.42 ± 1.05	26.77 ± 2.27 #	20.94 ± 2.87 *	23.59 ± 4.16
7-KLCA	7-ketolithocho lic acid	0.010 ± 0.01	0.013 ± 0.00	0.011 ± 0.01	0.009 ± 0.009
HDCA	Hyodeoxycho lic acid	1.40 ± 0.27	1.37 ± 0.41	1.43 ± 0.50	2.02 ± 0.35 *
LCA	Lithocho lic acid	0.05 ± 0.01	0.07 ± 0.01 #	0.04 ± 0.00 **	0.05 ± 0.02
CDCA	Chenodeoxycho lic acid	0.85 ± 0.07	1.24 ± 0.12 ##	1.60 ± 0.11 **	1.49 ± 0.17 **
3β-CA	3β-Cho lic Acid	0.01 ± 0.00	0.013 ± 0.00	0.008 ± 0.00 *	0.006 ± 0.00 **
CA	Cho lic acid	7.02 ± 3.80	2.76 ± 1.61	25.80 ± 26.68 *	3.26 ± 2.68
DCA	Deoxycho lic acid	0.96 ± 0.43	0.20 ± 0.14 ##	0.46 ± 0.46	0.31 ± 0.20
Tω-MCA	Tauro-ω-muricho lic Acid sodium salt	15.58 ± 3.63	22.31 ± 5.28 #	14.59 ± 2.93 *	18.81 ± 1.20
TUDCA	Tauroursodeoxycho lic acid	10.77 ± 4.90	14.20 ± 1.88	9.80 ± 3.63	7.14 ± 1.19 *
TLCA	Taurolithocho lic acid	0.11 ± 0.02	0.16 ± 0.02 ##	0.14 ± 0.02	0.13 ± 0.01 *
TDCA	Taurodeoxycho lic acid	19.82 ± 4.65	11.37 ± 5.33	23.27 ± 9.57 *	15.16 ± 6.29
TCDCA	Taurochenodeoxycho lic acid	4.70 ± 1.37	7.68 ± 1.73 #	8.10 ± 1.61	4.64 ± 1.66 *
TCA	Taurocholic acid	122.82 ± 22.07	90.77 ± 24.94 #	124.96 ± 16.91 *	81.42 ± 16.28
THDCA	Taurohyodeoxycho lic Acid sodium salt	8.47 ± 1.95	12.42 ± 2.30 #	6.22 ± 3.02 **	5.53 ± 2.68 *
Tβ-MCA	Tauro-β-muricholic acid	0.42 ± 0.14	0.92 ± 0.20 ##	0.51 ± 0.22 **	0.46 ± 0.12 *

# statistically significant difference in mean values between HFD and Control groups, # *p* < 0.05, ## *p* < 0.01. * statistically significant difference in mean values between the HFD group and the other two intervention groups: the HFD + Isorhamnetin group and the HFD + Simvastatin group.* *p* < 0.05, ** *p* < 0.01.

## Data Availability

The original contributions presented in the study are included in the article, further inquiries can be directed to the corresponding author.

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
