# Peer review of "Isorhamnetin in Quinoa Whole-Grain Flavonoids Intervenes in Non-Alcoholic Fatty Liver Disease by Modulating Bile Acid Metabolism through Regulation of FXR Expression"

_foods, 2024, doi:10.3390/foods13193076_

Round 1
Reviewer 1 Report
Comments and Suggestions for Authors
Section 2.3. Provide full detail of the identification and quantification parameters.
Section 2.6. Justify the selection of the treatment concentrations, as well as the selection of those three flavones.
Section 2.10.1. Justify the selection of isorhamnetin.
Section 2.10.5. Provide full details of the LC-MS conditions, as well as the identification and quantification parameters.
Section 2.11. Provide full detail of the antibodies.
Section 2.12. Describe how the authors assessed that all variables were parametric.
Table 1. Several MS/MS fragments correspond to the molecular ion [M+H]+, please provide full identification parameters since many polyphenols have the same molecular formula. Why isorhamnetin galactoside eluted after the aglycone? The chromatographic elution gradient suggest otherwise. How authors identified isomers (isorhamnetin vs rhamnetin or galactoside vs glucoside) and the position of conjugates (3-galactoside, 3-rutinoside, 4-hydroxy, 5-methyl)? Why authors focused only in flavonoids and not in other phenolic components? The selection of isorhamnetin, a low abundant flavonoid in quinoa, is not completely justify. If the selection was going to be carried out based on their potential bioactivity and commercial standards were going to be assessed in the in vitro and in vivo models, why characterize quinoa?
Figures 1C-1H are not clearly described nor easily observed in the manuscript. Consider having figures 1A-B as supplementary material.
Figure 2 is difficult to observe and is poorly described. Figure 3C does not include statistical analysis.
Authors must provide full-detail on how bile acids were identified and quantified (Table 2). Why authors focused in assessing hypocholesterolemic effects in a NAFLD model, when other metabolic alterations are also observed? like triglycerides accumulation, the main clinical characteristic of this disease.
Discussion must deepen in why only isorhamnetin exerted a beneficial effect and not the other flavones. Moreover, the profile of bile acids is not discussed, then why was it measured if only genes were going to be used for understanding the mechanism. A brief limitation statement must be included at the end of this section.
Author Response
Comments 1: Section 2.3. Provide full detail of the identification and quantification parameters.
Response 1: Thank you for your attention. The detailed process and related parameters of the UPLC-ESI-MS/MS Experiment have been described below for your review. In addition, we have added the CAS numbers of the reagents to the corresponding positions in the manuscript and put Table S1 and Table S2 and the relevant experimental parameters of the UPLC-ESI-MS/MS experiment into the Supplementary File.
Experimental reagents and instruments
|
Name |
CAS |
Purity |
Company |
|
Methanol |
67-56-1 |
LC-MS grade |
Merck KGaA |
|
Acetonitrile |
75-05-8 |
LC-MS grade |
Merck KGaA |
|
Fomic acid |
64-18-6 |
LC-MS grade |
Merck KGaA |
|
UHPLC |
Thermo Vanquish UHPLC |
- |
Thermo Fisher Scientific |
|
HRMS |
Q-Exactive HF |
- |
Thermo Fisher Scientific |
|
Chromatographic column |
Zorbax Eclipse C18 (1.8μm*2.1*100mm) |
- |
Agilent technologies |
Table S1 Liquid chromatography mobile phase conditions
|
Time (min) |
Flow rate (μL/min) |
Gradient |
B% Acetonitrile |
|
0-2 |
300 |
- |
5 |
|
2-6 |
300 |
Linear gradient |
30 |
|
6-7 |
300 |
- |
30 |
|
7-12 |
300 |
Linear gradient |
78 |
|
12-14 |
300 |
- |
78 |
|
14-17 |
300 |
Linear gradient |
95 |
|
17-20 |
300 |
- |
95 |
|
20-21 |
300 |
Linear gradient |
5 |
|
21-25 |
300 |
- |
5 |
Positive Mode: Heater Temperature 325°C; Sheath Gas Flow Rate: 45 arb; Auxiliary Gas Flow Rate: 15 arb; Purging Gas Flow Rate: 1 arb; Electrospray Voltage: 3.5 KV; Capillary Temperature: 330 °C; S-Lens RF Level: 55%.
Negative Mode: Heater Temperature 325 °C; Sheath Gas Flow Rate: 45 arb; Auxiliary Gas Flow Rate: 15 arb; Purge Gas Flow Rate: 1 arb; Electrospray Voltage: 3.5 KV; Capillary Temperature: 330 °C; S-Lens RF Level: 55%.
Scanning modes: Full Scan (m/z 100~1500) with data-dependent secondary mass spectrometry (dd-MS2, TopN = 10); Resolution: 120,000 (primary mass spectrometry) & 60,000 (secondary mass spectrometry). Collision mode: High energy collisional dissociation (HCD).
The components in CQWF30 were subsequently identified by primary MS and 25 potentially bioactive flavonoids in CQWF30 were further identified by secondary MS. The following table shows the 25 active flavonoids identified by MS in CQWF30.
Table S2 The flavonoids composition in CQWF30.
|
No. |
RT [min] |
Formula |
Class of compound |
Name |
[M+H]+/[M-H]- |
MW |
Percentage of CQWF30 |
|
1 |
6.20 |
C33H40O19 |
Flavanoids |
Mauritianin |
739.21 |
740 |
14.56 |
|
2 |
6.68 |
C15H10O7 |
Flavonoids |
Quercetin |
301.04 |
302 |
10.54 |
|
3 |
6.11 |
C32H38O20 |
Flavanoids |
Helicianeoide B |
741.19 |
742 |
9.28 |
|
4 |
6.23 |
C15H10O6 |
Flavonoids |
Kaempferol |
287.23 |
286 |
8.83 |
|
5 |
6.70 |
C21H20O12 |
Flavanoids |
Isoquercitrin |
463.09 |
464 |
8.66 |
|
6 |
6.82 |
C21H18O13 |
Flavanoids |
Quercetin3-O-β-D-Glucuronide |
477.07 |
478 |
7.9 |
|
7 |
6.27 |
C27H30O15 |
Flavanoids |
Nictoflorin |
595.52 |
594 |
7 |
|
8 |
6.05 |
C27H30O16 |
Flavanoids |
Quercetin3-O-glucoside-7-O-rhamnoside |
609.15 |
610 |
6.79 |
|
9 |
6.47 |
C27H30O16 |
Flavonoids |
Rutin |
611.52 |
610 |
3.84 |
|
10 |
6.29 |
C34H42O20 |
Flavonoids |
Typhaneoside |
771.69 |
770 |
3.43 |
|
11 |
7.07 |
C21H18O12 |
Flavanoids |
Scutellarin |
463.37 |
462 |
2.73 |
|
12 |
7.07 |
C21H18O12 |
Flavanoids |
Kaempferol-3-glucuronide |
461.07 |
462 |
2.41 |
|
13 |
6.36 |
C32H38O19 |
Flavanoids |
Camelliaside B |
725.19 |
726 |
2.32 |
|
No. |
RT [min] |
Formula |
Class of compound |
Name |
[M+H]+/[M-H]- |
MW |
Percentage of CQWF30 |
|
14 |
8.04 |
C14H22O |
Flavanoids |
4-Octylphenol |
205.46 |
206 |
2.22 |
|
15 |
6.37 |
C26H28O16 |
Flavanoids |
Quercetin 3-O-β-D-ribosyl-(1-2)-β-D-glucoside |
595.13 |
596 |
2.18 |
|
16 |
6.01 |
C21H20O12 |
Flavanoids |
Hyperoside |
465.38 |
464 |
2.15 |
|
17 |
6.25 |
C21H20O11 |
Flavanoids |
Orientin |
449.38 |
448 |
1.91 |
|
18 |
6.52 |
C28H32O16 |
Flavonoids |
Narcissin |
623.16 |
624 |
1.34 |
|
19 |
6.30 |
C16H12O7 |
Flavanoids |
Isorhamnetin |
317.26 |
316 |
0.82 |
|
20 |
6.88 |
C28H32O16 |
Flavonoids |
Isoscoparin-2"-β-D-glucopyranoside |
623.16 |
624 |
0.71 |
|
21 |
7.12 |
C22H22O12 |
Flavanoids |
Isorhamnetin3-galactoside |
477.10 |
478 |
0.24 |
|
22 |
2.97 |
C8H7NO2 |
Flavanoids |
4-Hydroxymandelonitrile |
150.15 |
149 |
0.08 |
|
23 |
9.13 |
C16H12O5 |
Flavanoids |
Glycitein |
283.06 |
284 |
0.08 |
|
24 |
5.23 |
C27H30O17 |
Flavanoids |
Baimaside |
625.14 |
626 |
0.05 |
|
25 |
9.13 |
C16H12O5 |
Flavanoids |
5-O-Methylgenistein |
285.26 |
284 |
0.01 |
Comments 2: Section 2.6. Justify the selection of the treatment concentrations, as well as the selection of those three flavones.
Response 2: Thanks for your question. Following MS analysis, we identified that CQWF30 contains 25 flavonoids including kaempferol, quercetin, and isorhamnetin, as detailed in Response 1 Table 3. Through preliminary screening of these flavonoids for relevance to NAFLD, we identified 13 flavonoids that may have an active effect on NAFLD (Table 1 and Section 3.1). Further analysis using network pharmacology suggested that kaempferol, quercetin, and isorhamnetin could be effective components with potential intervention effects on NAFLD (Figure 1 and Section 3.1).
Based on this result, we utilized the CCK-8 assay to evaluate the cytotoxicity of the three flavonoids above—kaempferol, quercetin, and isorhamnetin—on the hepatocyte cell lines HepG2 and BEL-7402. The selected treatment concentrations were 0, 10, 20, and 40 μM and the treatment time was 24 hours. Given the previous observation in the oil red O staining assay of CQWF30 that a reduction in intracellular lipid accumulation has been shown at a concentration of 30 μM (DOl: 10.1002/jsfa.13923), we chose concentrations close to this range to test for cytotoxicity in this experiment.
HepG2 BEL-7402
Comments 3: Section 2.10.1. Justify the selection of isorhamnetin.
Response 3: Thanks for your attention. As mentioned in section 3.3, isorhamnetin, a flavonoid compound, is a natural metabolic byproduct of quercetin, generated through metabolic processes within the organism [1,2]. It has been shown that ingested quercetin undergoes metabolism by gut microbiota, converting to quercetin-3-β-D-glucoside through interaction with water. This is then hydrolyzed by β-glucosidase, ultimately yielding isorhamnetin, with a physiological conversion efficiency as high as 80% [1]. Additionally, quercetin can also be metabolized into isorhamnetin in the liver. Given this information, coupled with the greater economic viability of quercetin compared to isorhamnetin, we opted to use quercetin instead of isorhamnetin to test its efficacy in interventions involving mice with NAFLD.
References:
- Tanaka, S.; Trakooncharoenvit, A.; Nishikawa, M.; Ikushiro, S.; Hara, H. Comprehensive Analyses of Quercetin Conjugates by LC/MS/MS Revealed That Isorhamnetin-7- O-glucuronide-4'- O-sulfate Is a Major Metabolite in Plasma of Rats Fed with Quercetin Glucosides. J Agric Food Chem 2019, 67, 4240-4249.
- Kim, M.; Jee, S.C.; Kim, K.S.; Kim, H.S.; Yu, K.N.; Sung, J.S. Quercetin and Isorhamnetin Attenuate Benzo[a] pyrene-Induced Toxicity by Modulating Detoxification Enzymes through the AhR and NRF2 Signaling Pathways. Antioxidants (Basel) 2021, 10,787.
Comments 4: Section 2.10.5. Provide full details of the LC-MS conditions, as well as the identification and quantification parameters.
Response 4: Thanks for your attention.
The main instruments used in this experiment were an LC-20AD HPLC (Shimadzu Corporation) and an ABSCIEX4000Q TRAP mass spectrometer (Scientific Export).
In addition, a total of 20 bile acid controls were available in this experiment for parametric comparison of the identified bile acids.
Take mouse liver tissue, add activated charcoal to remove endogenous bile acids, homogenize and stir, centrifuge, and take the supernatant as the blank matrix. The rest of the tissue was homogenized, centrifuged and the supernatant was taken. 50 μL was sucked up precisely, 200 μL of the internal standard solution was added, vortexed for 10 min, centrifuged at 12 000 r/min for 10 min, 200 μL of the supernatant was taken, blown dry under nitrogen gas, reconstituted with 100 μL of methanol, vortexed for 3 min, and then injected into the sample for analysis.
The separation was performed on a Waters Symmetry C18 column (2.1 mm×150 mm, 3.5 μm) with the mobile phase A as an aqueous solution containing 0.1% formic acid and 10 mmol/L ammonium acetate, and the mobile phase B as a methanol solution containing 0.1% formic acid and 10 mmol/L ammonium acetate, and the gradient elution (0~2 min, 60% B; 2~40 min. 60%B→90%B; 40~45 min, 90%B; 45~50 min, 60%B) at a flow rate of 0.15 mL/min with an injection volume of 10 μL, column temperature of 40℃, and autosampler: 4℃.
The mass spectrometry conditions were electrospray ionization (ESI), negative ion mode (-), and multiple ion reaction monitoring (MRM). The working parameters were as follows: air curtain gas was 137.9 kPa; ion source voltage was -4.5 kV; ion source temperature was 300℃; GS1 was 275.8 kPa; GS2 was 275.8 kPa; Interface Heater: on; EP was -10 V; and CXP was -13 V. The ESI was operated in the negative ion mode (-), and MRM was performed in the negative ion mode (-).
Each bile acid control was weighed precisely, placed in a 100 mL measuring flask, and dissolved with methanol to configure the standard control. The internal standard d4-GCDCA was weighed precisely, dissolved with acetonitrile, fixed, and prepared into a solution with a concentration of 5.0 μmol/L (acetonitrile was used as a protein precipitant).
The sample was analyzed by the mass spectrometer, and the bile acids detected were as shown in Table 2.
In addition, we have included this section in the Supplementary File as a detailed guide to the LC-MS/MS Experiment.
Comments 5: Section 2.11. Provide full detail of the antibodies.
Response 5: Thank you for your require. The antibodies mentioned in Section 2.11, as stated in Section 2.1, were all provided by Wuhan Mitaka Biotechnology. The table below provides the specific details of each antibody for your review. Additionally, we changed lines 262-264 of the text from “and specific antibodies were applied to label the target proteins.” to “and specific antibodies FXR, BSEP, SLCO1B3, and GAPDH were applied to label the target proteins respectively.” and highlighted in red. In addition, we have added the corresponding CAS numbers of the antibodies to the corresponding sections in the manuscript.
|
Name |
CAS |
Reactivity |
Company |
|
FXR/NR1H4 Polyclonal antibody |
25055-1-AP |
Human, Mouse, Rat, Pig |
Wuhan Mitaka Biotechnology |
|
BSEP Monoclonal antibody |
67512-1-Ig |
Human, Mouse, Rat, Pig |
|
|
SLCO1B3/OATP1B3 Monoclonal antibody |
66381-1-Ig |
Human |
|
|
HRP-conjugated Goat Anti-Rabbit IgG (H+L) |
SA00001-2 |
Rabbit |
|
|
HRP-conjugated Goat Anti-Mouse IgG (H+L) |
SA00001-1 |
Mouse |
|
|
GAPDH Polyclonal antibody |
10494-1-AP |
Human, Mouse, Rat, Pig, Arabidopsis, Cabbage, Rice |
Comments 6: Section 2.12. Describe how the authors assessed that all variables were parametric.
Response 6: Thanks for your attention. To make sure that all variables used in our data were parametric, we assessed the normal distribution (Gaussian distribution) of these variables and found that the variance was homogeneous between the groups being compared. For example, the body weights of the mice within each group conformed to a normal distribution. Meanwhile, when qPCR experiments were performed to detect the expression of various genes such as FXR, we performed three repetitions of independent experiments to ensure the independence of the results. And, for the data in the experiments, Z-value analysis was used to identify potential outliers and remove them.
Comments 7: Table 1. Several MS/MS fragments correspond to the molecular ion [M+H]+, please provide full identification parameters since many polyphenols have the same molecular formula. Why isorhamnetin galactoside eluted after the aglycone? The chromatographic elution gradient suggest otherwise. How authors identified isomers (isorhamnetin vs rhamnetin or galactoside vs glucoside) and the position of conjugates (3-galactoside, 3-rutinoside, 4-hydroxy, 5-methyl)? Why authors focused only in flavonoids and not in other phenolic components? The selection of isorhamnetin, a low abundant flavonoid in quinoa, is not completely justify. If the selection was going to be carried out based on their potential bioactivity and commercial standards were going to be assessed in the in vitro and in vivo models, why characterize quinoa?
Response 7: Thanks for your constructive suggestions. The [M+H]+ values for the relevant molecules have been attached to the Table S2 in Comments 1 for your review.
In addition, isorhamnetin galactoside is a glycoside analogue that contains a sugar portion in its molecule, which may increase its overall polarity and molecular weight, leading to its stronger interaction with the stationary phase in the column, and therefore moves slower during elution and subsequently elutes from the glycoside isorhamnetin. Furthermore, because there may be variations in each experimental run, this could result in interactions or adsorption between isorhamnetin and galactoside during the sample preparation process, thus impacting their elution sequence.
Secondly, in mass spectrometry, two or more compounds having the same molecular weight can be distinguished by their unique mass spectral features. The identity of a compound can first be confirmed by looking at the isotopic distribution pattern of the ion peaks of a molecule, which can vary from compound to compound. To further identify the compounds, we select specific ions based on primary mass spectra and perform secondary mass spectral fragmentation and detection to distinguish compounds with very similar structures, which is our main method of distinguishing these compounds.
We focused on flavonoids because after we identified CQWF30 by mass spectrometry, we found that the proportion of flavonoid content (~160 mg/ml) was significantly higher than that of polyphenols (~50 mg/ml), and plant flavonoids have also been reported by various parties to have biological activities that can have better intervention effects on a variety of chronic diseases in human beings, so we chose flavonoids as the study object.
From our mass spectrometry results, the percentage of isorhamnetin was only 0.82%, however, we chose isorhamnetin initially due to the network pharmacology screening that isorhamnetin, as a bioactive flavonoid, has the potential to intervene in NAFLD. As for the specific reference to isorhamnetin in quinoa, firstly, because even though the same substance extracted from different kinds of plants theoretically has similar functions, its presence in different plants may result in differences in extraction methods, which leads to differences in the functions of the extracted substance to a certain extent, and secondly, characterizing quinoa, a crop that has the potential to intervene in NAFLD, through our study, will be an important step for the development of quinoa. application to provide theoretical support.
Comments 8: Figures 1C-1H are not clearly described nor easily observed in the manuscript. Consider having figures 1A-B as supplementary material
Response 8: Thank you for your reminder, your comment is useful, and we have made the original Figure 1A-B as supplementary data Figure S1A-B and adjusted the other images of the original Figure 1 to make them more clearly visible and corrected the numbering of the photo references in the text, with the changes marked in red.
And, to better describe Figure 1, we have made some changes to the Results 3.1 section, changing the text to read “The study conducted an in-depth exploration of 53 potential targets through GO enrichment analysis. Firstly, by analyzing the biological processes associated with these targets, we found that they primarily participate in reactions involving inorganic substances and peptides, positive regulation of phosphorylation processes, and regulation of hormone levels (Fig. 1C). In the Cellular Component (CC) analysis, it was evident that these targets show significant associations with functional areas such as membrane rafts, vesicle lumens, extracellular matrix, glutamatergic synapses, and RNA polymerase II transcription regulatory complexes (Fig. 1D). Furthermore, enrichment analysis of genes related to molecular functions revealed a close relationship between these targets and enzymatic activities such as oxidoreductase activity and nitric oxide synthase regulatory activity (Fig. 1E).
Additional pathway analysis using the KEGG highlighted the involvement of these targets in multiple disease-related metabolic pathways. These pathways encompass cancer signaling pathways relevant to tumorigenesis and development, lipid metabolism and atherosclerosis pathways that demonstrate the link between abnormal lipid metabolism and cardiovascular diseases, as well as bile acid metabolism pathways (Fig. 1F). Given the intimate connection between lipid metabolism and bile acid metabolism, the latter being a critical process for regulating hepatic lipid excretion and maintaining bile flow, its dysfunction has been linked to NAFLD [21]. Therefore, it is plausible that the bioactive molecules in CQWF30—kaempferol, quercetin, and isorhamnetin—may intervene in NAFLD by influencing the bile acid metabolism pathway.” In lines 293-312.
Comments 9: Figure 2 is difficult to observe and is poorly described. Figure 3C does not include statistical analysis.
Response 9: Thanks for your constructive suggestions. We have adjusted some of the content of the Results 3.2 section, which now reads “To delve deeper into the effects of these compounds on alleviating NAFLD, we established a high-lipid cellular model induced by FFA. In HepG2 cells, the FFA-induced model group exhibited a significantly increased accumulation of lipid droplets (Fig. 2G-I). However, after treatment with quercetin and isorhamnetin, there was an observed reduction in intracellular lipid droplets (Fig. 2H-I). In contrast, kaempferol treatment did not bring about any notable changes (Fig. 2G). Similar observations were corroborated in BEL-7402 cells (Fig. 2J-L).
Subsequently, Nile red fluorescence staining conducted in HepG2 cells demonstrated that both quercetin and isorhamnetin could effectively reduce the accumulation of lipid droplets induced by fatty acids, with isorhamnetin showing particularly significant results, consistent with oil red O staining (Fig. 2M-2O). The Nile red staining experiment in BEL-7402 cells also highlighted the superior effect of isorhamnetin in reducing lipid accumulation (Fig. 2P-2R). Based on these findings, we chose to focus subsequent research efforts on isorhamnetin. We performed quantitative analyses of TG and T-CHO in the FFA-induced NAFLD cellular models. Results indicated that isorhamnetin could effectively inhibit the increase in T-CHO and TG levels induced by FFA in HepG2 cells (Fig. 2S-2T), a phenomenon also observed in BEL-7402 cells (Fig. 2U-2V).
Taken together, our study demonstrated that isorhamnetin can inhibit the accumulation of lipids in high-lipid cells induced by FFA.” in lines 329-346.
In addition, we have analyzed the data in Figure 3C for differences and have represented them in the figure below.
Comments 10: Authors must provide full-detail on how bile acids were identified and quantified (Table 2). Why authors focused in assessing hypocholesterolemic effects in a NAFLD model, when other metabolic alterations are also observed? like triglycerides accumulation, the main clinical characteristic of this disease.
Response 10: Thanks for your constructive suggestions. We identified bile acids in mouse liver by LC-MS/MS technique. For the accuracy of the identification, we used various bile acid standards as a reference. The data (peak times, etc.) obtained from the samples by mass spectrometry were compared with those of the bile acid standards to determine the type of bile acids as mentioned in Comments 4. The quantification of bile acids is based on the peak areas of different bile acids.
In addition, we observed a reduction in lipid accumulation within HepG2 and BEL-7402 in hepatocytes after isorhamnetin intervention, and our study was designed to decipher the mechanism of this effect of isorhamnetin. Since excess lipids in hepatocytes of healthy individuals are usually eliminated from the body by bile acid metabolism, the balance of bile acid metabolism is a major factor in maintaining the balance of lipid metabolism in hepatocytes. However, the sediment of bile acids is a major clinical phenomenon in patients with NAFLD, and therefore, we speculated whether isorhamnetin achieves the reduction of intracellular lipid accumulation in hepatocytes by regulating bile acid metabolism. In conclusion, we first found that isorhamnetin intervention can reduce lipid accumulation in fatty liver cells, then considered that bile acid metabolism balance can maintain lipid metabolism homeostasis, and finally focused on the metabolic homeostatic effect of isorhamnetin on bile acids.
Comments 11: Discussion must deepen in why only isorhamnetin exerted a beneficial effect and not the other flavones. Moreover, the profile of bile acids is not discussed, then why was it measured if only genes were going to be used for understanding the mechanism. A brief limitation statement must be included at the end of this section.
Response 11: Thanks for your constructive suggestions. Based on network pharmacology screening, our research predicted that kaempferol, quercetin, and isorhamnetin, three flavonoids, may have intervention effects on NAFLD. Experimental results indicate that both quercetin and isorhamnetin can inhibit lipid accumulation, with isorhamnetin showing particularly significant effects. Considering that isorhamnetin is a major metabolite of quercetin, we believe that isorhamnetin might be the key substance inhibiting lipid accumulation. We speculate that the unique chemical structure of isorhamnetin (a 3'-O-methylated flavonol backbone) could make it more stable or easier to absorb than other flavonoids, which explains its superior performance in reducing lipid accumulation.
Further studies revealed that isorhamnetin effectively regulates bile acid transport and reabsorption in mouse models of nonalcoholic fatty liver disease, significantly lowering TBA levels while increasing CDCA levels and markedly decreasing levels of UDCA and T-β-MCA. CDCA is the strongest endogenous agonist for FXR, whereas UDCA and T-β-MCA are antagonists of FXR. This finding suggests that isorhamnetin enhances FXR expression activity by modulating bile acid composition, elevating CDCA levels, and reducing UDCA and T-β-MCA levels. Activated FXR further promotes the expression of BSEP while inhibiting the expression of SLCO1B3, alleviating symptoms of bile acid stasis and thus mitigating the progression of NAFLD.
Therefore, isorhamnetin’s intervention primarily works by altering bile acid composition, subsequently activating or suppressing the expression of genes related to bile acid metabolism, achieving a balanced state of bile acid metabolism, and ultimately alleviating NAFLD symptoms. Therefore, besides studying gene expression, changes in bile acid levels should also be considered.
It is worth noting that although isorhamnetin has shown positive effects, this does not rule out the roles of other flavonoids. Given that flavonoids in nature often act synergistically, we need further exploration into the combined effects of multiple flavonoids. To this end, we have added relevant content to the discussion section.
We changed “Our findings uncovered that isorhamnetin proficiently regulated bile acid transport and reabsorption in a NAFLD mouse model, effectively lowering TBA con-tent, markedly elevating CDCA levels, and considerably decreasing UDCA and T-β-MCA levels.” to “We understand that in patients with NAFLD, compared to healthy individuals, the ratio of primary to secondary BAs within the liver can shift, leading to an increased proportion of secondary BAs and a concomitant rise in total bile acid levels, a phenomenon known as bile acid retention. Additionally, the signaling pathways involving BAs may be disrupted, preventing the effective activation of receptors such as FXR, which are crucial for metabolic regulation. In our study, the administration of iso-rhamnetin significantly reduced the levels of bile acids such as α-MCA, UDCA, ω-MCA, LCA, Tω-MCA, THDCA, and Tβ-MCA in the livers of NAFLD model mice, while concurrently increasing the concentrations of CDCA and TCA (Fig. 5).” in line 513-521 and add “through modulating the composition of BAs” in line 527, also “Furthermore, what we have currently observed is that isorhamnetin intervention modulates the bile acid composition within the liver of a NAFLD model; however, the underlying mechanisms of this phenomenon remain unclear. Additionally, it is yet to be determined whether other flavonoids in CQWF30 or the synergistic effects of multiple flavonoid compounds also exert similar interventions on NAFLD. These questions will be the focus of our future research efforts.” In lines 533-538. All the changes have been highlighted in red for review.

Reviewer 2 Report
Comments and Suggestions for Authors
General comments
The revised manuscript “Isorhamnetin in Quinoa Whole-Grain Flavonoids Intervenes in Non-Alcoholic Fatty Liver Disease by Modulating Bile Acid Metabolism through Regulation of FXR Expression” by Xiaoqin La et al. is an original and well presented manuscript. The work explores in silico, in vitro and in vivo the beneficial effects of isorhamnetin, presented as component of CQWF30, to dietary interventions on NAFLD.
The introduction is appropriate and relevant. The aim of the manuscript is clear and original. The experiments are appropriate and well presented.
Despite of these, the manuscript did not evidence how the effects described to isorhamnetin alone are representative of those effects attributed to CQWF30. In this respect, authors cited a M.D. thesis (cite 20) which is not available on line to analyze the effects reported previously on CQWF30.
It is a well-recognized fact that botanical preparations, as CQWF30, have biological effects related to the additive, synergistic or antagonic interactions between its phytochemical components. In this respect would be desirable that author include in this manuscript the effects induced by the CQWF30 to the easy comparison with those induced by isorhamnetin alone on the in vitro and in vivo assays.
Resuming, the manuscript is a very important contribution to elucidate the beneficial effects of isorhamnetin on NAFLD. However, there are necessary major changes to correlate the isorhamnetin activity with the CQWF30 actions before the manuscript be published. This reviewer strongly encourage the authors to include the missing information and resubmit the manuscript.
Some particular comments to improve a future presentation are listed below.
Particular comments
Line 20: please, change “In vitro” to italics.
Line 22: please, change “In vivo” to italics.
Line 49: please, explain the acronyms the first time that they appears. I.e. “BAs”.
Lines 85 to 87: “Our previous study high lighted that CQWF30, isolated and purified from quinoa whole-grain, has a remarkable hepatoprotective effect in alleviating NAFLD, nonetheless, the precise identity of the core bioactive constituent remained undetermined”. Which are the previous studies that the authors cite?
In line 133. “A 100 mg sample of CQWF30 was taken…” How are the 100 mg of sample quantified? What does the value represent? …the estimated grain weight in solution or the suspended lyophilized preparation. Please, include a clear sentence indicating how the weight of CQWF30 used in the assays is calculated.
Line 186. The “protocol” mentioned… Which is the protocol used related to the extraction solution? Please, clarify the procedure indicating the solution composition and the protocol followed. Same to protocol mentioned in line 192. Are they the manufacturer’s protocols?
Line 222. Why the animals included in the isorhamnetin group received quercetin? Please, correct or clarify the sentence.
Line 239. Which is the standard solution added to the liver tissue? Please, correct or clarify.
Line 271. Cite 20 it is not available as valid precedent to confirm the activity of CQWF30 on NAFLD mitigation. Readers can not access to the previous studies that support the current manuscript.
Please, improve visualization of figure 1. As it is, letters in the figure are not readable.
Line 306. Cytotoxicity of kaempferol, quercetin and isorhamnetin are tested in the manuscript. Because of the results, it is clear that isolated compounds do not affect cell viability. However, the experimental design under-estimate the potential synergistic activity between the probed compounds, when they are present in CQWF30 sample. Because of this, in the cytotoxicity studies, as well as in the following assays, the treatment with different doses of CQWF30 as control is strongly suggested.
In figure 2, please improve the presentation of statistical differences in panels S, T, U and V.
Author Response
General comments
The revised manuscript “Isorhamnetin in Quinoa Whole-Grain Flavonoids Intervenes in Non-Alcoholic Fatty Liver Disease by Modulating Bile Acid Metabolism through Regulation of FXR Expression” by Xiaoqin La et al. is an original and well presented manuscript. The work explores in silico, in vitro and in vivo the beneficial effects of isorhamnetin, presented as component of CQWF30, to dietary interventions on NAFLD.
The introduction is appropriate and relevant. The aim of the manuscript is clear and original. The experiments are appropriate and well presented.
Despite of these, the manuscript did not evidence how the effects described to isorhamnetin alone are representative of those effects attributed to CQWF30. In this respect, authors cited a M.D. thesis (cite 20) which is not available on line to analyze the effects reported previously on CQWF30.
It is a well-recognized fact that botanical preparations, as CQWF30, have biological effects related to the additive, synergistic or antagonic interactions between its phytochemical components. In this respect would be desirable that author include in this manuscript the effects induced by the CQWF30 to the easy comparison with those induced by isorhamnetin alone on the in vitro and in vivo assays.
Resuming, the manuscript is a very important contribution to elucidate the beneficial effects of isorhamnetin on NAFLD. However, there are necessary major changes to correlate the isorhamnetin activity with the CQWF30 actions before the manuscript be published. This reviewer strongly encourage the authors to include the missing information and resubmit the manuscript.
Some particular comments to improve a future presentation are listed below.
Response to the General comments
First of all, thank you for your kind recognition of our work. Regarding the concern that the manuscript did not describe whether the effects of isorhamnetin can represent those of CQWF30, we offer the following clarification: The master’s thesis originally cited (Reference 20) was not available online, which caused inconvenience to you; for this, we extend our sincere apologies. To address this issue, we have replaced the reference with one that can be accessed via DOI: 10.1002/jsfa.13923. The research in this article (DOI: 10.1002/jsfa.13923) is our preliminary work, and the article elaborates on the extraction, preparation, and purification process of CQWF30 and discusses its role in the intervention of NAFLD. It is worth noting that CQWF30 contains multiple components, and at higher concentrations such as 30 μM, it may lead to partial cell death. In contrast, isorhamnetin exhibits lower toxicity to liver cells during the intervention and demonstrates a better effect in reducing intracellular lipid accumulation.
Currently, the research related to the extraction and purification of CQWF30 and the intervention of NAFLD effects has been submitted to the Journal of the Science of Food and Agriculture and has been accepted by the journal. You can access it via DOI: 10.1002/jsfa.13923. For each question you have asked, we have attached the corresponding figure to the question for you to review. If you can’t find it using this DOI number, you can access it on this link.
(https://d.wanfangdata.com.cn/thesis/ChJUaGVzaXNOZXdTMjAyNDAxMDkSCUQwMjcwNDM1MBoIZm5ndmR3bzU%3D)
Response to the Particular comments
Comments 1: Line 20: please, change “In vitro” to italics.
Response 1: Thank you for your reminder. We have changed “In vitro” in line 20 to italics and highlighted it in red for your viewing.
Comments 2: Line 22: please, change “In vivo” to italics.
Response 2: Thanks for your reminder again. We have changed “In vivo” in line 22 to italics and highlighted it in red for your viewing.
Comments 3: Line 49: please, explain the acronyms the first time that they appears. I.e. “BAs”.
Response 3: Thank you for your suggestion. We have changed “Studies have shown that BAs” to “Studies have shown that Bile Acids (BAs)” in line 49 of the text and highlighted it in red for you review.
Comments 4: Lines 85 to 87: “Our previous study high lighted that CQWF30, isolated and purified from quinoa whole-grain, has a remarkable hepatoprotective effect in alleviating NAFLD, nonetheless, the precise identity of the core bioactive constituent remained undetermined”. Which are the previous studies that the authors cite?
Response 4: Thank you for the reminder. The reference to the hepatoprotective effect of CQWF30 in lines 85-87 of the text is the result of our previous research, which you can access via DOI: 10.1002/jsfa.13923. In addition, according to your suggestion, we have added the citation number [20] here.
Comments 5: In line 133. “A 100 mg sample of CQWF30 was taken…” How are the 100 mg of sample quantified? What does the value represent? …the estimated grain weight in solution or the suspended lyophilized preparation. Please, include a clear sentence indicating how the weight of CQWF30 used in the assays is calculated.
Response 5: Thanks for the question. The CQWF30 mentioned in line 133 of the text was obtained by ultrasound-assisted ethanol extraction. We first extracted the crude flavonoid extract from the quinoa whole-grain, and subsequently, different concentrations of ethanol were used to elute various components from the CQWF. The eluted solutions were then freeze-dried to produce a dry powder. Therefore, the reference to 100 mg of CQWF30 in the text refers to the weighing of 100 mg of freeze-dried CQWF30 powder. The detailed experimental procedure can be referred to the extraction and purification method of CQWF30 described in the literature (DOI: 10.1002/jsfa.13923).
Comments 6: Line 186. The “protocol” mentioned… Which is the protocol used related to the extraction solution? Please, clarify the procedure indicating the solution composition and the protocol followed. Same to protocol mentioned in line 192. Are they the manufacturer’s protocols?
Response 6: Thanks for your question. The “protocol” mentioned in lines 186 and 192 refers to the procedure in the instruction manual of the TG assay kit used for TG content determination.
The TG assay kit we used was purchased from Beijing Solarbio Technology Co., Ltd (CAS: BC0625)and the following are the main steps of the protocol.
|
Reagent name |
Specification |
|
Reagent 1 |
Self-contained reagents |
|
Reagent 2 |
10 Ml×1 |
|
Reagent 3 |
15Ml×1 |
|
Reagent 4 |
15Ml×1 |
|
Reagent 5 |
15Ml×1 |
|
Reagent 6 |
15Ml×1 |
|
Standard sample |
Powder |
Determination principle: TG was extracted with isopropyl alcohol, KOH saponified TG and hydrolyzed to produce glycerol and fatty acids, periodic acid oxidation of glycerol to produce formaldehyde, in the presence of chlorine ions, formaldehyde, and acetylacetone condensed to produce a yellow substance, with characteristic absorption peaks at 420 nm, and the depth of the color is proportional to the content of TG.
Preparation of solutions:
1, Reagent 1: Prepare your empty glass bottle, n-heptane and isopropanol mixed according to the volume ratio of 1:1, tightly capped and mixed, about 150mL, ready to use, ready to mix. Keep it at 2-8℃;
2, standard sample: Before use, add 5 mL of reagent 1, that is, 1 mg/mL triglyceride standard solution, -20 ℃, stored for a fortnight, avoiding repeated freezing and thawing.
Operation Steps:
Sample processing
1, Tissue: according to the mass of tissue (g): reagent one volume (mL) of 1:5~10 ratio (it is recommended to weigh about 0.1g of tissue, add 1mL of reagent one) for ice bath homogenization, 8000g, centrifugation at 4℃ for 10min, take the supernatant to be measured.
2, Cells, bacteria: first collect 500-1000 million cells or bacteria into the centrifuge tube, centrifuge and discard the supernatant, add 1mL of reagent 1, ultrasonic breakage for 1min (power 200W, ultrasonic 2s, stop 1s), centrifuge at 8000g and 4℃ for 10min, and take the supernatant to be measured.
3, Serum (plasma) and other liquid samples: direct measurement.
Measurement steps
- Preheat the spectrophotometer/enzymometer for 30 min, adjust the wavelength to 420 nm, and zero with distilled water.
- Preheat the water bath to 65℃.
|
|
BLANK |
STANDARD |
TEST |
|
Standard liquid (μL) |
- |
120 |
- |
|
TG test liquid (μL) |
- |
- |
120 |
|
Reagent 1 (μL) |
495 |
375 |
375 |
|
Reagent 2 (μL) |
75 |
75 |
75 |
Add reagent 1 and mix thoroughly, then add reagent 2, shake vigorously for 30s, let it stand for 3-5min, then shake vigorously for 30s, and so on for 3 times, and let it stand for a certain time at room temperature.
- Determination of TG content:
|
|
BLANK |
STANDARD |
TEST |
|
Upper solution (μL) |
30 |
30 |
30 |
|
Reagent 3 (μL) |
100 |
100 |
100 |
|
Reagent 4 (μL) |
30 |
30 |
30 |
|
Mix thoroughly, 65 ℃ water bath for 3min, cooling |
|||
|
Reagent 5 (μL) |
100 |
100 |
100 |
|
Reagent 6 (μL) |
100 |
100 |
100 |
|
Mix thoroughly, 65 ℃ water bath for 15min, cooling |
|||
After cooling, pipette 200 μL into a micro glass cuvette/96-well plate to measure the absorbance at 420 nm and record as A blank, A standard, and A test. (Blank and standard tubes should only be tested 1-2 times)
- TG Calculation Formula
- Serum (plasma) triglyceride level:
TG level (mg/dL) = C standard × (A test - A blank) ÷ (A standard - A blank) × 100 = 100 × (A test - A blank) ÷ (A standard - A blank)
- Triglyceride content in tissues or cells, bacteria:
(1) Calculated from sample protein concentration
TG content (mg/ mg prot) = C standard × V × (A test - A blank) ÷ (A standard - A blank) ÷ (Cpr × V) = (A test - A blank) ÷ (A standard - A blank) ÷ Cpr
(2) Calculated by sample mass
TG content (mg/ g mass) = C standard × V × (A test - A blank) ÷ (A standard - A blank) ÷ W = (A test - A blank) ÷ (A standard - A blank) ÷ W
(3) Calculation of TG content by cell/bacteria count (mg/104 cell) = C standard × (A test - A blank) ÷ (A standard - A blank) × V ÷ N=(A test - A blank) ÷ (A standard - A blank) ÷N
C standard: 1mg/mL; 100: unit conversion factor, 1dL=100mL; Cpr: sample protein concentration, mg/mL; W: sample mass, g; V: Volume of reagent 1 added, 1mL; N: Number of bacteria or cells, in tens of thousands.
Comments 7: Line 222. Why the animals included in the isorhamnetin group received quercetin? Please, correct or clarify the sentence.
Response 7: Thank you for your attention. As stated in Section 3.3, isorhamnetin, a natural metabolite of quercetin, is formed during the metabolic processes within the organism. After quercetin is ingested, it is metabolized by the gut microbiota and reacts with water to form quercetin-3-β-D-glucoside. Subsequently, under the action of β-glucosidase, it undergoes further hydrolysis, ultimately yielding isorhamnetin, with this conversion process being up to 80% efficient. Additionally, quercetin can also be metabolized into isorhamnetin in the liver. Considering this, along with the higher economic feasibility of quercetin compared to isorhamnetin, we opted to use quercetin to test its efficacy in treating a mouse model of non-alcoholic fatty liver disease. This experimental strategy is also supported by previous research, as demonstrated in our reference [24].
Comments 8: Line 239. Which is the standard solution added to the liver tissue? Please, correct or clarify.
Response 8: Thank you for your attention. In the determination of BAs in the liver of mice, we utilized HPLC-MS/MS technology to measure the types and levels of BAs present in different groups. Initially, we added an internal standard solution of Glycochenodeoxycholic Acid-d4 (d4-GCDCA) to the collected liver tissues. This solution was prepared by dissolving d4-GCDCA powder in acetonitrile to achieve a concentration of 5.0 μmol·L-1 (acetonitrile served as a protein precipitation agent). And we have changed line 244 of the text from “standard solution” to “standard solution (Glycochenodeoxycholic Acid-d4)”.
Comments 9: Line 271. Cite 20 it is not available as valid precedent to confirm the activity of CQWF30 on NAFLD mitigation. Readers can not access to the previous studies that support the current manuscript.
Response 9: Thank you for the reminder and we apologize for the lack of support for citation [20]. You can find information about CQWF30 intervention in NAFLD at DOI: 10.1002/jsfa.13923.
Comments 10: Please, improve visualization of figure 1. As it is, letters in the figure are not readable.
Response 10: Thank you for your suggestion. We have adjusted Figure 1 for clarity and the letters in the figure are now clearly visible, in response to Reviewer 1’s comments, we have placed figures A and B from the original Figure 1 in the Supplementary Data as Figure S1 A and B. The re-adjusted Figure 1 is shown below for your review.
Figure 1
Comments 11: Line 306. Cytotoxicity of kaempferol, quercetin and isorhamnetin are tested in the manuscript. Because of the results, it is clear that isolated compounds do not affect cell viability. However, the experimental design under-estimate the potential synergistic activity between the probed compounds, when they are present in CQWF30 sample. Because of this, in the cytotoxicity studies, as well as in the following assays, the treatment with different doses of CQWF30 as control is strongly suggested.
Response 11: We appreciate your constructive suggestions. Through the CCK-8 assay, we evaluated the cytotoxicity of kaempferol, quercetin, and isorhamnetin. Our findings indicate that these substances do not exhibit any significant toxic side effects on cells. However, we observed more pronounced toxic phenomena in the oil red O and BODIPY assays with HepG2 and BEL-7402 cells treated with high concentrations of CQWF30. This toxicity might result from synergistic effects from these three compounds or other toxic components present in CQWF30. These observations further support our conclusion that isorhamnetin, one of the compounds we have identified in CQWF30, can intervene in NAFLD without causing severe cellular toxicity, making it a promising intervention approach. Oil red O experiment and BODIPY on the toxic effects of CQWF30 on cells
fluorescence staining assay, you can refer to DOI: 10.1002/jsfa.13923.
HepG2 BEL-7402
Comments 12: In figure 2, please improve the presentation of statistical differences in panels S, T, U and V.
Response 12: Thank you for your helpful suggestion, we have changed the presentation of statistical differences in S, T, U, and V in Figure 2 to the following figure for your review.
Figure 2

Round 2
Reviewer 2 Report
Comments and Suggestions for Authors
With exception of the cite [20], all the general and particular comments were adressed by the authors, in the manuscript and/or the cover letter.
I suggest to avance with manuscript publication after this minor change were solved.
Author Response
CommentWith exception of the cite [20], all the general and particular comments were adressed by the authors, in the manuscript and/or the cover letter.
I suggest to avance with manuscript publication after this minor change were solved.
Response: We appreciate your valuable suggestions. Regarding the issue of the unavailability online of reference [20] cited in our manuscript, we would like to provide the following explanation: This reference documents the early research outcomes of our research group, which has been accepted for publication by the Journal of the Science of Food and Agriculture (DOI: 10.1002/jsfa.13923). The initial proofreading was completed on September 20, 2024. Now, we are very pleased to inform you that our article has been accepted and successfully published online (as shown below). We would like to extend our sincere gratitude for your support and patience throughout this process. Thank you for your understanding and support.
